# Two RNA-binding proteins mediate the sorting of miR223 from mitochondria into exosomes

**Liang Ma, Jasleen Singh, Randy Schekman***

Department of Molecular and Cell Biology, Howard Hughes Medical Institute, University of California, Berkeley, United States

**Abstract** Fusion of multivesicular bodies (MVBs) with the plasma membrane results in the secretion of intraluminal vesicles (ILVs), or exosomes. The sorting of one exosomal cargo RNA, miR223, is facilitated by the RNA-binding protein, YBX1 (Shurtleff et al., 2016). We found that miR223 specifically binds a 'cold shock' domain (CSD) of YBX1 through a 5' proximal sequence motif UCAGU that may represent a binding site or structural feature required for sorting. Prior to sorting into exosomes, most of the cytoplasmic miR223 resides in mitochondria. An RNA-binding protein localized to the mitochondrial matrix, YBAP1, appears to serve as a negative regulator of miR223 enrichment into exosomes. miR223 levels decreased in the mitochondria and increased in exosomes after loss of YBAP1. We observed YBX1 shuttle between mitochondria and endosomes in live cells. YBX1 also partitions into P body granules in the cytoplasm (Liu et al., 2021). We propose a model in which miR223 and likely other miRNAs are stored in mitochondria and are then mobilized by YBX1 to cytoplasmic phase condensate granules for capture into invaginations in the endosome that give rise to exosomes.

## Editor's evaluation

This important study presents a novel mechanism of miRNA223 sorting into exosomes involving its storage within mitochondria, specifically by a mitochondrially localized protein YBAP1. The evidence supporting the findings is convincing and opens avenues for future studies on molecular mechanisms. This paper is a valuable addition to the cellular sorting of miRNA involving interplay with and between the organelles, interesting for miRNAs researchers, as well as cell biologists.

## Introduction

Extracellular vesicles (EVs) bud from the plasma membrane or are secreted when multivesicular bodies (MVB) fuse with the plasma membrane to release a population of vesicles called exosomes. EVs and their cargos are highly dependent on their membrane source. Microvesicles released by budding from the plasma membrane are a heterogeneous population of EVs ranging in size from 30 nm to 1000 nm (*Cocucci et al., 2009*). Exosomes are 30 nm to 150 nm in size and originate as vesicles invaginated into the interior of an MVB to form intraluminal vesicles (ILVs; *Harding et al., 1983*).

Many RNAs are selectively sorted into EVs, especially small RNAs. Several studies have indicated that RNA binding proteins (RNPs) may be involved in the enrichment of RNAs into EVs (*Mukherjee et al., 2016*; *Santangelo et al., 2016*; *Teng et al., 2017*; *Villarroya-Beltri et al., 2013*). However, many of these studies used sedimentation at ~100,000 g to collect EVs, which may also collect RNP particles not enclosed within membranes which complicates the interpretation of these data. To address this question, we previously developed buoyant density-based methods to separate EVs

**\*For correspondence:**
schekman@berkeley.edu

**Competing interest:** The authors declare that no competing interests exist.

from non-vesicular aggregates and found that EVs form two distinct populations of high and low buoyant density (*Shurtleff et al., 2016*; *Temoche-Diaz et al., 2020*). We found that some miRNAs are selectively enriched in a high buoyant density vesicle fraction characterized by an enrichment in the exosomal marker protein CD63, whereas the low buoyant density EVs are fairly non-selective in the capture of miRNAs (*Temoche-Diaz et al., 2019*). We developed a cell-free reaction to identify YBX1 as required for miR223 sorting into exosomes and demonstrated that it plays an important role in the enrichment of miR223 into exosomes in HEK293T cells (*Shurtleff et al., 2016*). We subsequently found that phase separated YBX1 condensates selectively recruit miR223 in vitro and sort it into exosomes in cells (*Liu et al., 2021*). In this study, we report that YBX1 directly and specifically binds miR223 by its 'cold shock' domain (CSD). We have identified a sequence motif, UCAGU, that facilitates the sorting of miR223 into exosomes. We also found a significant fraction of cytoplasmic miR223 localized within mitochondria, tightly associated with the mitochondrial envelope and that a mitochondrial RNA-binding protein, YBAP1, may control the transfer of miR223 from mitochondria to exosomes.

## Results

### YBX1 directly and specifically binds miR223

We previously documented that YBX1 facilitates miR223 sorting into exosomes (*Shurtleff et al., 2016*) and that exosomal miR223 is decreased in YBX1 knockout cells (*Liu et al., 2021*). We reexamined the enrichment and confirmed that exosomal miR223 was decreased in exosomes purified from YBX1 KO cells (*Figure 1a*). We used a Nanosight particle tracking device to quantify buoyant density purified vesicles and found that knockout of YBX1 did not affect exosome secretion (*Figure 1—figure supplement 1*).

Whereas the importance of YBX1 for miR223 sorting has been established, the mechanism of their interaction was not known. To examine the direct interaction of YBX1 and miR223, we used an electrophoretic mobility shift assay (EMSA) with purified recombinant YBX1, expressed in insect cells (*Figure 1—figure supplement 2a*), and chemically synthetic miR223 and miR190, a cytoplasmic miRNA that is not enriched in exosomes. Purified YBX1 was titrated and incubated with 5' fluorescently labeled miR223 at 30 °C for 30 min. miR223-YBX1 complexes were separated by electrophoresis and detected by in-gel fluorescence. The EMSA data showed that YBX1 directly and specifically bound to miR223, but ~140 fold less well with miR190 (*Figure 1b–c*). The measured Kd for YBX1:miR223 was 4.2 nM (*Figure 1d*).

YBX1 has three major domains including an N-terminal alanine/proline-rich (A/P) domain, a central cold shock domain (CSD) and a C-terminal domain (CTD) (*Figure 1e*). To explore which specific domain of YBX1 binds miR223, we constructed a series of fragments: the A/P domain, CSD and CTD. The YBX1 fragments were expressed in and purified from insect cells (*Figure 1—figure supplement 2b*). EMSA data showed that the A/P domain and CTD had little or no affinity for miR223, whereas the CSD domain bound miR223 but with an affinity much reduced compared to full length YBX1 (*Figure 1f*). We then constructed two combined fragments of the A/P and CSD and CSD and CTD domains (*Figure 1—figure supplement 2c*). The EMSA data showed that the A/P domain was dispensable, whereas binding of miR223 to CSD plus CTD was comparable to full-length YBX1 (*Figure 1g*).

YBX1-F85A in the CSD domain was reported to block the YBX1-specific binding of mRNA (*Lyons et al., 2016*). Purified YBX1-F85A protein failed to bind miR223 (*Figure 1g*, *Figure 1—figure supplement 2c*). These data suggest that YBX1 directly and specifically binds miR223 via the CSD. The CTD of YBX1 did not appear to bind miR223 but may somehow facilitate a higher affinity interaction of the CSD with miR223.

### A binding or structural motif on miR223 that promotes interaction with YBX1 and enrichment into exosomes

We next sought to determine the miR223 sequence motif responsible for interaction with YBX1 and enrichment into exosomes. We used an EMSA competition assay with a series of miR223 mutants. Purified YBX1 and 5' fluorescently labeled miR223 were incubated with miR223 mutant constructs titrated in a range from 1 nM to 1 µM. miR223 variants in a binding domain should not compete for interaction of YBX1 with 5' fluorescently tagged miR223 whereas variations in sequences irrelevant to

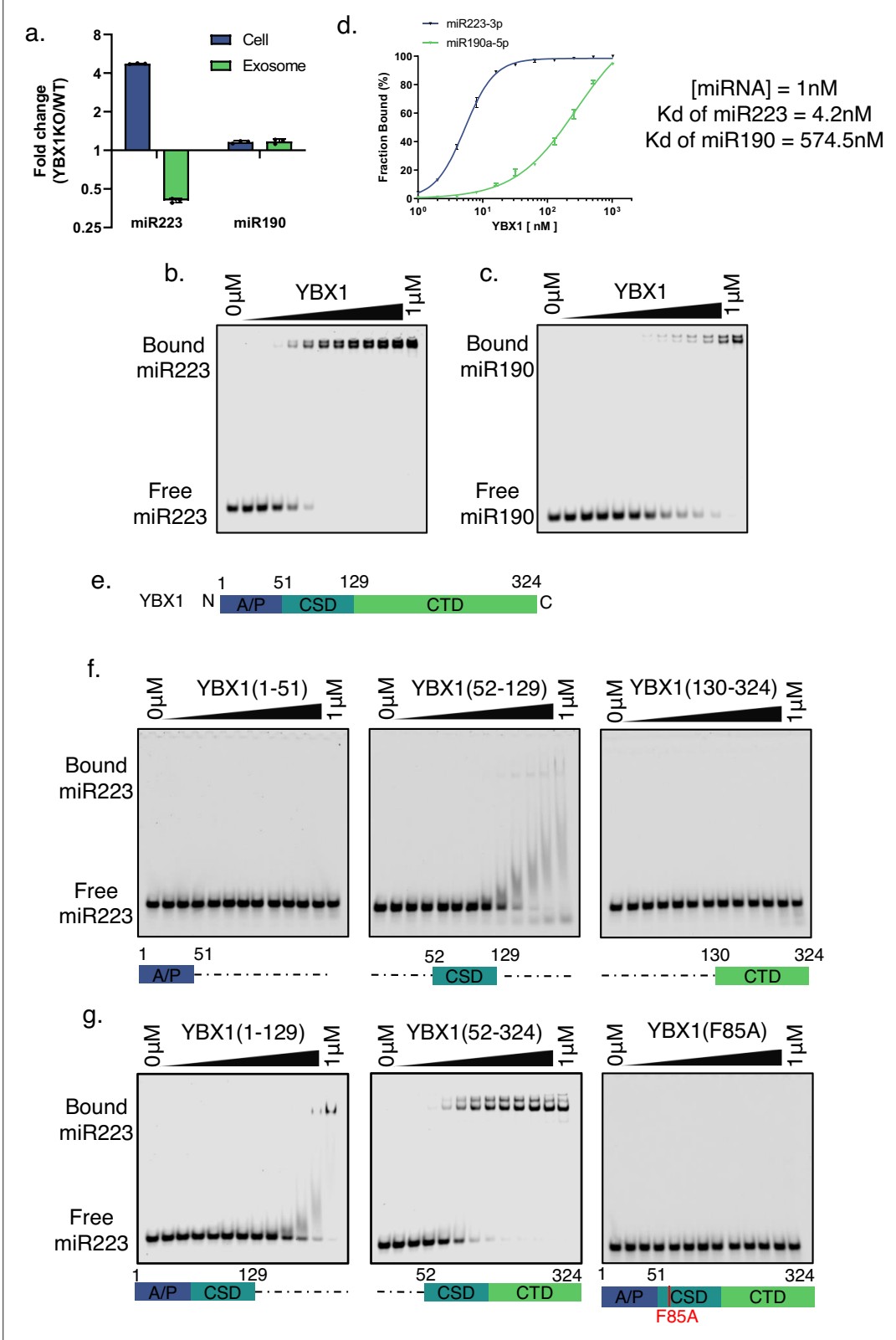

**Figure 1.** YBX1 directly and specifically binds miR223. (**a**) RT-qPCR analysis of fold change of miR-223 and miR-190 in cells and purified exosomes from 293 T WT cells and YBX1 knockout cells. Data are plotted from three independent experiments, each independent experiment with triplicate qPCR reactions; error bars represent standard deviations. (**b–c**) EMSA assays using 1 nM 5' fluorescently labeled miR223 or miR190 and purified YBX1.

*Figure 1 continued on next page*

*Figure 1 continued*

Purified YBX1 was titrated from 500pM to 1 μM. In gel fluorescence was detected. Quantification of (**d**) shows the calculated Kd. (**e**) Schematic diagrams of the different domains of YBX1. (**f**) EMSA assay using 1 nM 5′ fluorescently labeled miR223 and purified YBX1 truncations. [YBX1(1–51) or YBX1(52–129) or YBX1(130–324).] (**g**) EMSA assay using 1 nM 5′ fluorescently labeled miR223 and purified YBX1 truncations [YBX1(1–129) or YBX1(52–324)] or YBX1(F85A) mutant.

The online version of this article includes the following source data and figure supplement(s) for figure 1:

**Source data 1.** Uncropped gel images corresponding to *Figure 1*.

**Figure supplement 1.** Knockout of YBX1 did not change exosome secretion.

**Figure supplement 2.** Purified YBX1 full length protein and different truncations and mutation.

**Figure supplement 2—source data 1.** Uncropped gel images corresponding to *Figure 1—figure supplement 2*.

interaction would compete. Using this EMSA competition assay to screen the miR223 binding motif, we found that the competitive binding of miR223mut (3-6) and miR223mut (4-7) were decreased (*Figure 2—figure supplement 1*). This suggested that the sequence UCAGU was critical for interaction with YBX1. To test this directly, we employed a variant sequence, termed miR223mut, where the UCAGU was substituted with AGACA. As a positive control, we employed a variant of miR190, miR190sort, where the sequence AUAUG was substituted with UCAGU (*Figure 2a*). EMSA data showed a~27-fold reduced YBX1 interaction of with miR223mut, whereas the affinity of miR190sort with YBX1 was increased ~eightfold compared to wt sequences.

To test whether this motif is critical for miR223 enrichment into exosomes, we purified exosomes from 293T cells transiently transfected to overexpress one of the four miRNA constructs (*Figure 2d*). RT-qPCR data showed that the level of miR223 in exosomes was ~fourfold dependent on the putative exosomal sorting motif (*Figure 3e*) and the enrichment of miR190sort into exosomes was increased ~fivefold compared to miR190 WT (*Figure 2f*).

In previous work, we developed a cell-free reaction to test the biochemical requirement for YBX1 in the sorting of miR223 into vesicles formed with membranes and cytosol isolated from broken HEK293 cells (*Shurtleff et al., 2016*). In this work, we showed that the sorting of miR223 and of a CD63-luciferase fusion protein into an enclosed membrane were coincidentally inhibited by GW4869 an inhibitor of neutral sphingomyelinase (NS2) known to interfere with exosome biogenesis and secretion. On this basis, we concluded that the cell-free reaction recapitulated the sorting event leading to the packaging of miR223 into exosomes.

We refined this assay to measure the incorporation of $^{32}$P-5′ end-labeled wt and mutant miR223 into vesicles formed in vitro. Isolated membranes and cytosol were incubated with $^{32}$P-labled wt or mutant miR223 at 30 °C for 20 min, after which RNase I was added to digest any unpackaged miRNA. Controls including 1% Triton X-100 were used to measure background RNase resistant radiolabel. Samples were resolved on a gel for visual and quantitative evaluation of membrane sequestered RNA (*Figure 2g and h*). The results suggested that the UCAGU motif is critical for miR223 packaging into vesicles in the cell-free reaction.

Taken together the results in *Figure 2* show that the miR223 sequence UCAGA promotes the binding of YBX1 in order to sort the miRNA into vesicles formed in cells and in a cell-free reaction. We suggest this sorting facilitates the export of miR223 in exosomes secreted from HEK293 cells.

## Mitochondria contribute to miR223 enrichment into exosomes

In a recent study, we showed that YBX1 is sorted into P-bodies in cells and that these biomolecular condensates may initiate the sorting of miR223 into vesicles budding into the interior of endosomes (*Liu et al., 2021*; *Shurtleff et al., 2016*). Mitochondria represent another apparent intracellular location of miR223 (*Wang et al., 2020*). We used cell fractionation of homogenates of HEK293 cells to evaluate the subcellular distribution of endogenous miR223. Fractionation was evaluated by immunoblot using marker proteins characteristic of various cell organelles (*Figure 3a*). Analysis of RNA extracted from isolated membranous organelles confirmed that miR223 but not miR190 was significantly enriched in mitochondria but not in ER or cytosol (*Figure 3b*, *Figure 3—figure supplement 1a*; *Wang et al., 2020*).

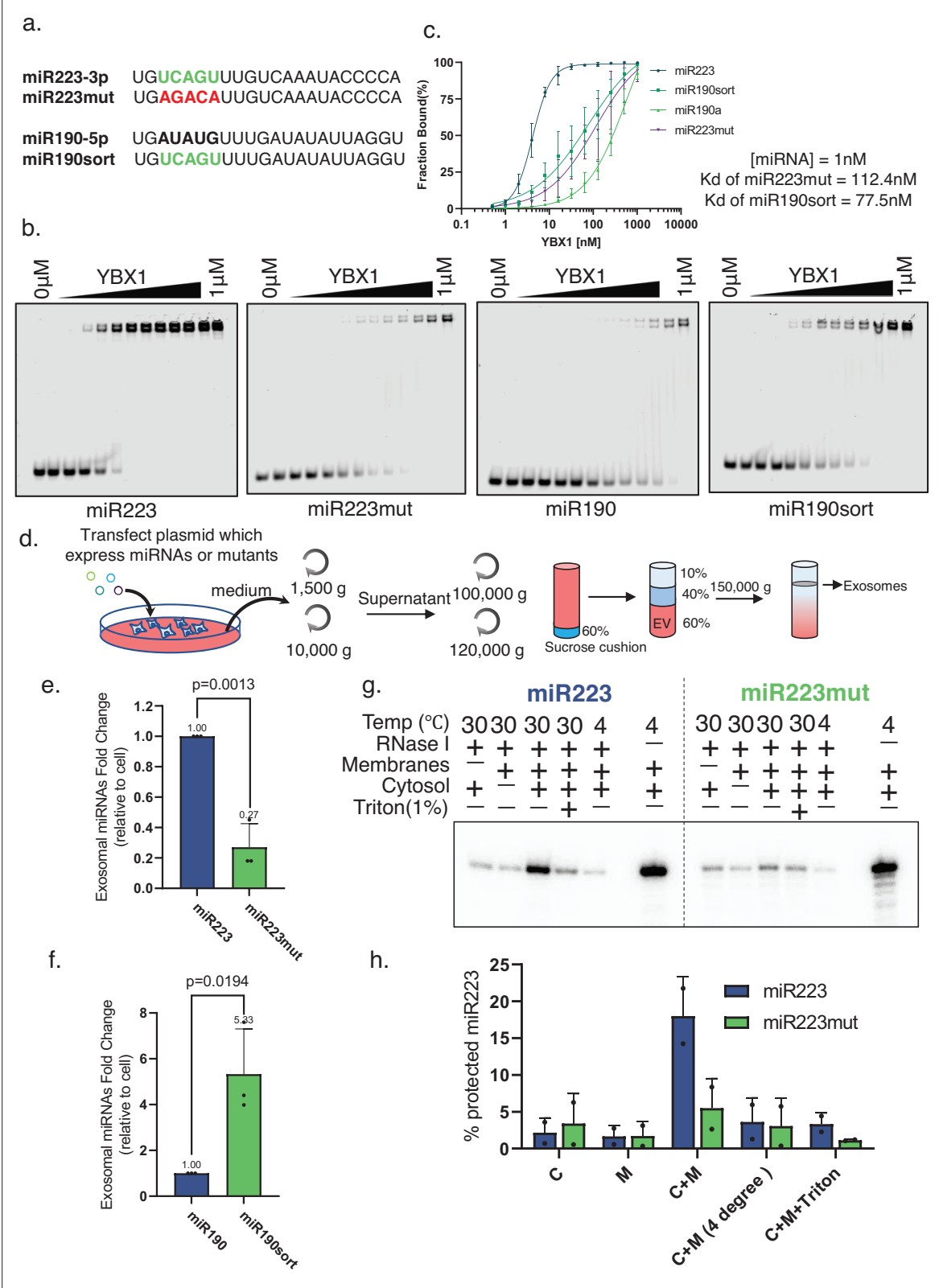

**Figure 2.** miR223 sequence motif UCAGU binds YBX1. (**a**) RNA oligonucleotides corresponding to miR223, miR190 and versions with mutated sorting motif (miR223mut) or mutation to introduce the sorting motif (miR190sort). (**b**) EMSA assays using 1 nM 5' fluorescently labeled miR223 WT or miR223mut or miR190 WT or miR190sort and purified YBX1. Purified YBX1 was titrated from 500pM to 1 µM. In gel fluorescence was detected. (**c**) Binding affinity curves as calculated by EMSA data from (**b**) (**d**) Schematic shows exosome purification with buoyant density flotation in a sucrose

*Figure 2 continued on next page*

*Figure 2 continued*

step gradient from 293T cells overexpressing miR223 WT or mutant or miR190 WT or miR190sort. (**e**) RT-qPCR analysis of relative abundance of miR223 or miR223mut detected in exosomes compared to cellular level in 293T cells overexpressing miR223 WT or miR223mut. Data are plotted from three independent experiments and error bars represent standard deviations. (**f**) RT-qPCR analysis of relative abundance of miR190 or miR190sort detected in exosomes compared to cellular level in 293T cells overexpressing miR190 WT or miR190sort. Data are plotted from three independent experiments and error bars represent standard derivations. (**g**) In vitro packaging assay using $^{32}$P 5'end-labeled miR223 and miR223mut. Cell-free packaging of miR223 and miR223mut measured as protected radioactive signal from $^{32}$P labeled miR223 and miR223mut. Reactions with or without membrane, cytosol, and 1% Triton X-100, and incubated at 4 or 30 °C are indicated. For the samples containing only cytosol plus membrane at 4 °C, only one-third of the samples were loaded. Each sample was supplemented with 300 mM urea to reduce the background signal. (**h**) Data quantification showed protected fraction of miR223 and miR223mut as calculated from in vitro packaging data shown in (**g**).

The online version of this article includes the following source data and figure supplement(s) for figure 2:

**Source data 1.** Uncropped gel images corresponding to *Figure 2*.

**Figure supplement 1.** Screening of exosomal sorting motif of miR223.

**Figure supplement 1—source data 1.** Uncropped gel images corresponding to *Figure 2—figure supplement 1*.

To determine the localization of miR223 on or within mitochondria, we prepared mitoplasts using digitonin to strip away the mitochondrial outer membrane followed by fractionation on a Percoll density gradient. Immunoblots of the enriched mitochondria and isolated mitoplasts showed that the outer membrane, marked by Tom20, was largely removed with retention of the inner membrane marker Tim23 (*Figure 3c*). RNA was extracted from the purified mitoplast and RT-qPCR data indicated that miR223 was enriched along with mRNA for COX1, but not with nuclear U6 snRNA (*Figure 3d*).

As an independent means to assess the localization of cytoplasmic miR223, we used immunoprecipitation to purify mitochondria. Isolated mitochondria were then converted to mitoplasts by osmotic shock and treated with proteinase K and RNase. Immunoblots of the immunoprecipitated mitochondria and isolated mitoplasts showed that the outer membrane, marked by Tom20, and intermembrane space, marked by AIF, were largely removed with retention of the mitochondrial matrix marker citrate synthase (*Figure 3e*). RNA was extracted from the immunoprecipitated mitochondria and mitoplasts and RT-qPCR data indicated that miR223 was enriched along with mRNA for COX1, but not with miR190 or nuclear U6 snRNA (*Figure 3f*).

We also used immunoprecipitated mitochondria (*Figure 3—figure supplement 1b*) and either Triton X-100 to solubilize the membrane or freeze-thaw to allow the matrix and envelope fractions to be separated by centrifugation. Mitochondrial membrane proteins, such as Tom20 and COX IV, were solubilized and retained in the supernatant fraction (*Figure 3—figure supplement 1c*). The freeze-thaw regimen released citrate synthase to a supernatant fraction whereas Tom20 and COX IV sedimented in the pellet fraction. RNA was extracted from the detergent supernatant and pellet fractions where we found similar distributions of COX1 and miR223, neither of which were as readily solubilized as the inner and outer membrane proteins (*Figure 3—figure supplement 1d*). RT-qPCR quantification of fractions from the freeze-thaw regimen showed that both COX1 mRNA and miR223 remained largely associated with the sedimentable membrane fraction (*Figure 3—figure supplement 1e*). We conclude that miR223 is enclosed within mitochondria, possibly in association with the inner membrane.

We sought a test of the role of mitochondria in the secretion of miR223 in exosomes. For this purpose, we generated cells depleted of mitochondria (*Correia-Melo et al., 2017*). U-2 OS cells expressing GFP-parkin were treated with CCCP for 48 hr, conditions that cause mitochondria to be removed by mitophagy. We confirmed mitochondrial depletion after CCCP treatment by RT-qPCR of mitochondrial COX1 mRNA (*Figure 3g*) and immunoblot of the mitochondrial inner membrane marker Tim23 (*Figure 3h*). We then compared the levels of both miR223 and miR190 from GFP-parkin expressing U-2 OS cells with and without CCCP treatment. RT-qPCR data showed that miR223, but not miR190, increased threefold in cells treated with CCCP (*Figure 3j*). To test the possibility that miR223 accumulated in cells as a result of a failure of mobilization into exosomes, we compared the miR223 levels in exosomes purified from untreated and CCCP treated cells (*Figure 3j*). Although exosome secretion, as measured with a CD63-luciferase marker, did not change after CCCP treatment (*Figure 3—figure supplement 2a*), we found that CCCP treatment lowered the amount of miR223 in EVs fourfold (*Figure 3j*). The increase in cellular at the expense of exosomal miR223 may reflect a critical role for mitochondria in the mobilization of this RNA to exosomes.

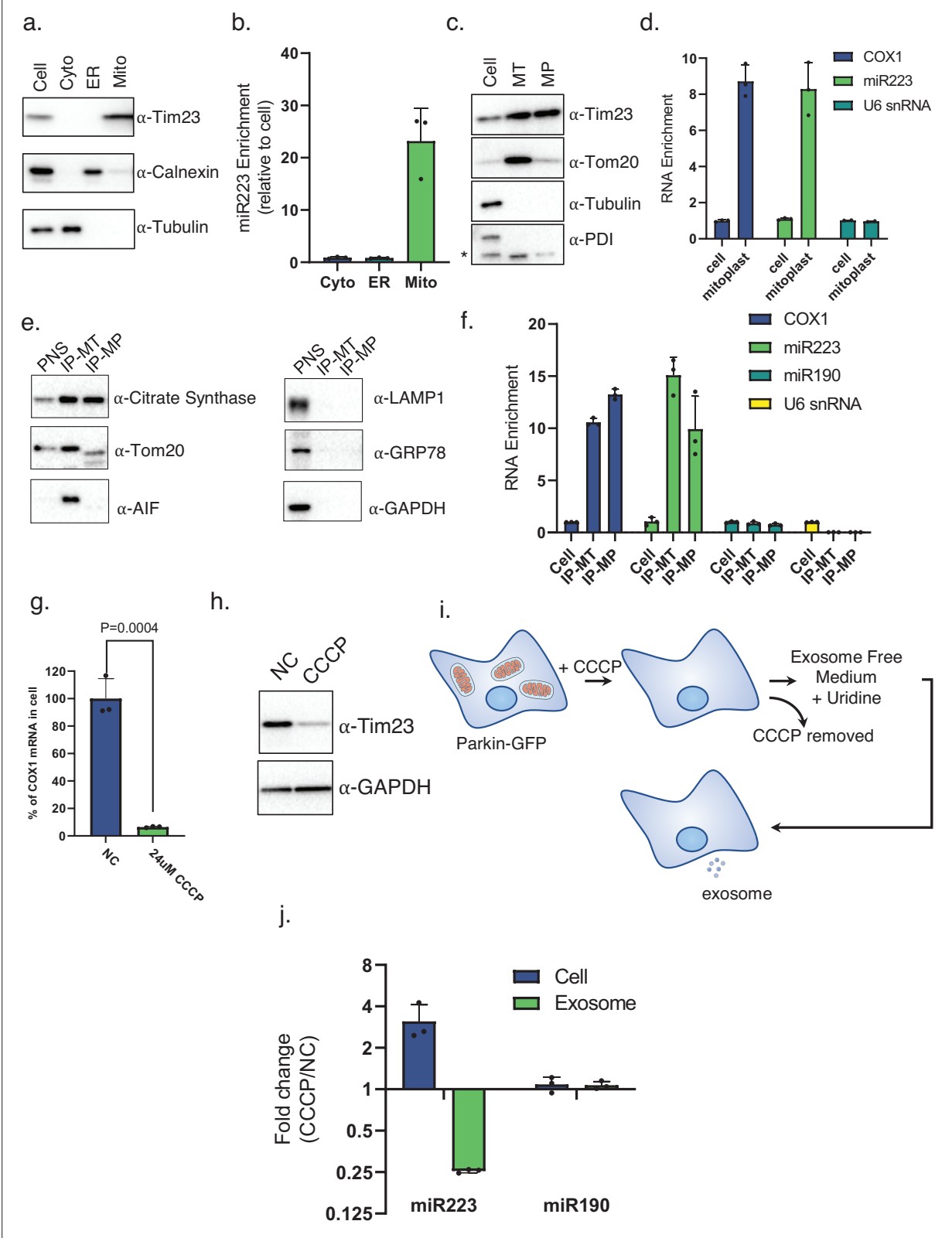

**Figure 3.** Mitochondria contribute to miR223 enrichment into exosomes. (**a**) Immunoblot analysis of protein markers for different subcellular fractions isolated from 293T cells. (**b**) RT-qPCR analysis of miR223 fold changes of different subcellular fractions isolated from 293T cells relative to cell lysate. (**c**) Immunoblot analysis of protein markers for mitoplasts purified from 293T cells by Percoll gradient fractionation (MT: mitochondria; MP: mitoplast). (**d**) RT-analysis of COX1 mRNA, miR223 and U6 snRNA fold changes for mitoplasts purified from 293T cells relative to cell lysate. (**e**) Immunoblot analysis

*Figure 3 continued on next page*

*Figure 3 continued*

of protein markers for immunoprecipitated mitochondria and osmotic shock generated mitoplasts. Mitochondria were purified from a 293T 3xHA-EGFP-OMP25 overexpressing cell line using anti-HA magnetic beads. Mitoplasts were purified following mitochondrial immunoprecipitation by osmotic shock, proteinase K and RNase treatment (IP-MT: immunoprecipitated mitochondria; IP-MP: immunoprecipitated mitoplasts). (**f**) RT-analysis of COX1 mRNA, miR223, miR190 and U6 snRNA fold changes for immunoprecipitaed mitochondria and mitoplasts purified from the 293T 3xHA-EGFP-OMP25 overexpressing cell line. Data are plotted from three independent experiments and error bars represent standard deviations. (**g**) RT-qPCR analysis of mitochondrial mRNA COX1 in U2OS cells expressing GFP-Parkin treated with or without CCCP. Data are plotted from three independent experiments and error bars represent standard deviations. (**h**) Immunoblot analysis of mitochondrial marker Tim23 in U2OS cells expressing GFP-Parkin treated with or without CCCP. (**i**) Schematic of exosome purification from mitochondria depleted GFP-Parkin expressing U2OS cells. (**j**) RT-qPCR analysis of fold change of miR-223 and miR-190 in cells and purified exosomes from U2OS cells expressing GFP-Parkin which were treated with or without CCCP. Data are plotted from three independent experiments and error bars represent standard deviations.

The online version of this article includes the following source data and figure supplement(s) for figure 3:

**Source data 1.** Uncropped immunoblot images corresponding to *Figure 3*.

**Figure supplement 1.** miR223, but not miR190, enriched in mitochondria.

**Figure supplement 1—source data 1.** Uncropped immunoblot images corresponding to *Figure 3—figure supplement 1*.

**Figure supplement 2.** Mitochondrial depletion did not change exosome secretion.

## YBAP1 binds miR223 in the mitochondria and in vitro

In the course of purifying a tagged version of YBX1 from 293T cells, we observed another protein that copurified and found that it corresponded to YBAP1 (*Figure 4a and b*). Such a complex of YBX1 and YBAP1 has previously been reported (*Matsumoto et al., 2005*). We confirmed that purified YBX1 and YBAP1 bind each other by coexpression and affinity purification from insect cells (*Figure 4—figure supplement 1b*). YBAP1 is a mitochondrial matrix protein with a standard N-terminal transit peptide sequence (*Muta et al., 1997*). We confirmed this mitochondrial localization in U-2 OS cells expressing Tom22-mCherry transiently transfected with a YBAP1-GFP construct (*Figure 4c–d*). We also showed that YBAP1 is localized within mitochondria by performing a proteinase K protection assay on purified mitochondria. Mitochondria were isolated from non-transfected cells and exposed to proteinase K in the presence or absence of Triton X-100 and the degradation of YBAP1 was evaluated by immunoblot. YBAP1 was resistant to proteinase K digestion as was the mitochondrial inner membrane marker Tim23. Both were degraded by proteinase treatment in the presence of Triton X-100 (*Figure 4e*), consistent with the localization of YBAP1 within mitochondria.

To test whether YBAP1 was bound to miR223 in mitochondria, we used YBAP1 immunoprecipitation with mitochondria purified by fractionation on a Percoll density gradient (*Figure 4f*). RT qPCR data showed that mitochondrial miR223 was immunoprecipitated by YBAP1 antibody but not by a control antibody (*Figure 4g*). To determine whether the YBAP1 and miR223 interaction was direct, we used the EMSA assay and found that purified YBAP1 bound miR223, but not miR190. The YBAP1 interaction with miR223 was not dependent on the RNA sequence motif responsible for YBX1 binding (*Figure 4—figure supplement 2a–b*). Taken together, these data suggest that YBAP1 binds miR223 in mitochondria and in vitro.

## YBAP1 may control the transit of miR223 from mitochondria to exosomes

To investigate the function of YBAP1 in the transit of miR223 into exosomes, we generated a 293T YBAP1 KO cell line and compared the level of miR223 enrichment in exosomes and mitochondria isolated from WT and mutant cells (*Figure 5a and b*). Although knockout of YBAP1 did not change exosome secretion (*Figure 5—figure supplement 1a*), RT-qPCR analysis showed that miR223 decreased twofold in mitochondria but increased eightfold in exosomes purified from mutant and WT cells, respectively (*Figure 5c*). This apparent inverse relationship is consistent with a role for YBAP1 protein in the retention of miR223 in mitochondria.

## YBX1 puncta shuttled from mitochondria to endosomes

In previous work we reported the localization of YBX1 to P-bodies and suggested this may represent an intermediate stage in the concentrative sorting of miRNAs for secretion in exosomes (*Liu et al., 2021*). In other earlier work, P-bodies were seen in association with mitochondria (*Huang et al.,*

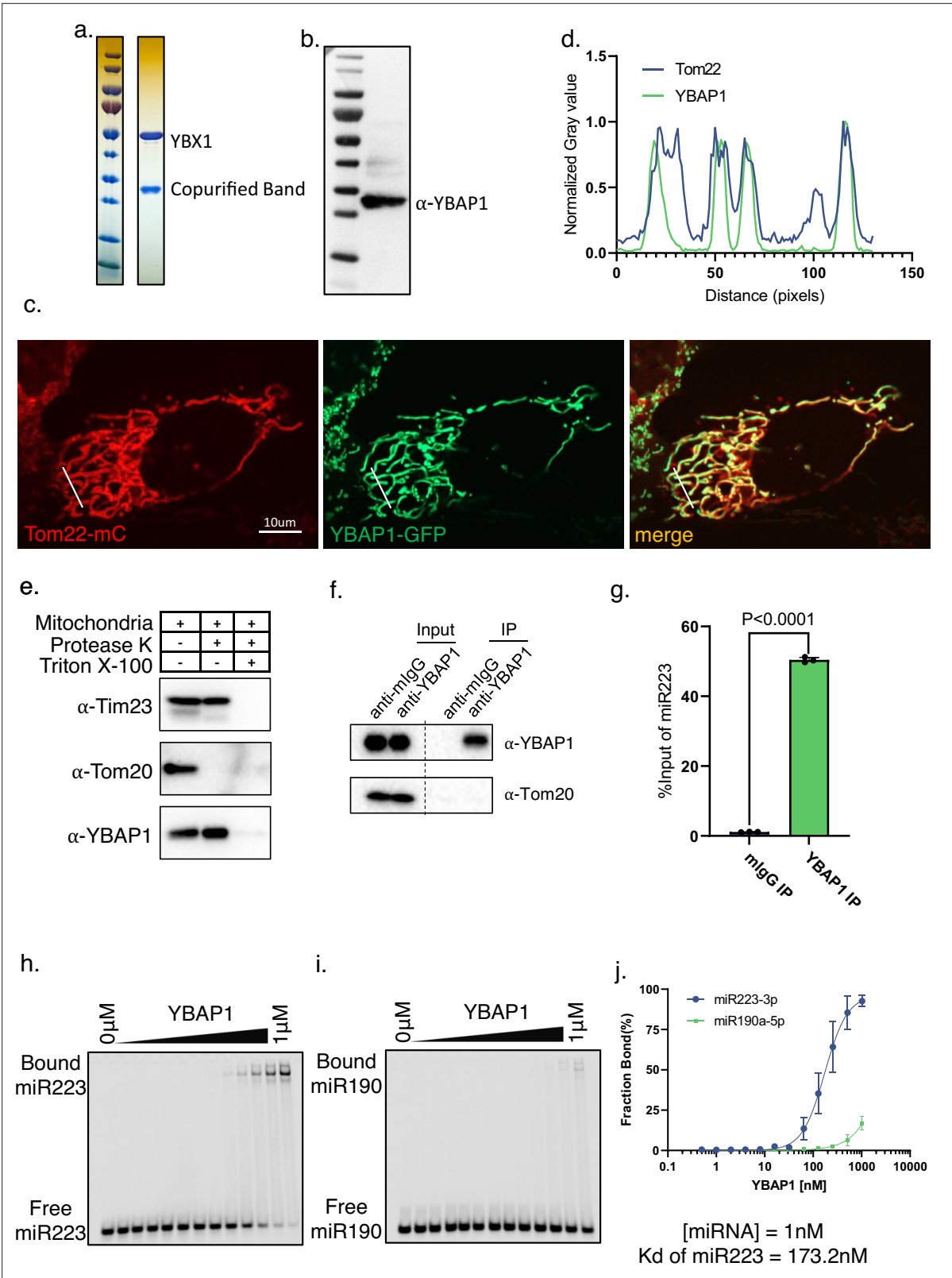

**Figure 4.** YBAP1 directly and specifically binds miR223. (**a**) Strep II-YBX1 was overexpressed in HEK293T cells. Coomassie blue detection of unknown band copurified with YBX1 from 293T cells. (**b**) Immunoblot identified unknown band was YBAP1. (**c**) Tom22-mCherry expressing U2OS was transfected with a YBAP1-GFP-expressing plasmid, cultured for 12 hr and observed by confocal microscopy. Scale bar, 10 μm. (**d**) Quantification of the fluorescence intensity of the different channels indicated by the solid white line of (**c**). (**e**) YBAP1 resides in mitochondria. Proteinase K protection assay for YBAP1

*Figure 4 continued on next page*

*Figure 4 continued*

using purified mitochondria from 293T cells. Samples were treated with or without proteinase K (10 µg/ml) and or Triton X-100 (0.5%). Immunoblots for Tim23, Tom20, and YBAP1 are shown. (**f**) Mitochondria were purified for immunoprecipitation with YBAP1 antibody. Immunoblot detection of YBAP1 and Tom20. (**g**) RT-qPCR analysis of miR223 fold changes of YBAP1 IP samples. Data are plotted from three independent experiments and error bars represent standard deviations. (**h–i**) EMSA assays using 1 nM 5' fluorescently labeled miR223 or miR190. Purified YBAP1 was titrated from 500pM to 1 µM. In gel fluorescence was detected. Quantification of (**j**) shown the calculated Kd.

The online version of this article includes the following source data and figure supplement(s) for figure 4:

**Source data 1.** Uncropped immunoblot and gel images corresponding to *Figure 4*.

**Figure supplement 1.** YBX1 and YBAP1 copurify as a complex from transfected SF9 cells.

**Figure supplement 1—source data 1.** Uncropped gel images corresponding to *Figure 4—figure supplement 1*.

**Figure supplement 2.** YBAP1 does not share the same miR223-binding motif as YBX1.

**Figure supplement 2—source data 1.** Uncropped gel images corresponding to *Figure 4—figure supplement 2*.

*2011*). To explore this possibility, we visualized endogenous YBX1 and YBAP1 by IF and observed YBX1 puncta colocalized with mitochondria (*Figure 6a and b*). In order to detect the proximity of endosomes to this point of contact between YBX1 puncta and mitochondria, we used U-2 OS cells transfected with Rab5(Q79L)-mCherry, which we employed previously to enlarge and detect the internalization of YBX1 into endosomes (*Liu et al., 2021*). We then used three color visualization of the U-2 OS cells also transfected with YFP-YBX1 and mito-BFP. Time-lapse imaging showed YBX1 puncta in close proximity to mitochondria or endosomes, followed quickly by transfer between them (*Figure 6c and d*). Taken together, these data suggest a mechanism whereby miR223 stored in mitochondria, possibly sequestered by YBAP1, may be captured in a tighter interaction with YBX1 in P-bodies and delivered to endosomes for sorting and secretion in exosomes.

## Discussion

Selected miRNAs are sorted, some with very high fidelity, into invaginations in the endosome that give rise to exosomes secreted from cultured human cells and likely from many if not all cells in metazoan organisms. The means by which these miRNAs are sorted and the possible extracellular functions they serve is a subject of interest in normal and disease physiology. Here we report the role of the RNA-binding protein YBX1 and a sorting or structural signal on one target RNA, miR223, and the indirect path miR223 may take from storage in mitochondria into exosomes.

We have identified a sequence motif on miR223, UCAGU, responsible for high-affinity interaction with YBX and for sorting into vesicles formed in a cell-free reaction as well as for secretion in exosomes by HEK293 cells. Previously we performed this in vitro packaging assay in the presence of an inhibitor (GW4869) of neutral sphingomyelinase (NS2). This inhibitor has been shown to reduce the secretion of exosomes and exosome-associated miRNAs in other studies (*Li et al., 2013*; *Trajkovic et al., 2008*; *Yuyama et al., 2012*). In our cell-free assay, GW4869 inhibited the protection of CD63-luciferase and miR-223 at concentrations known to inhibit the activity of NS2 in partially purified enzyme fractions (*Shurtleff et al., 2016*). We concluded that our cell-free reaction provides a model that mimics aspects of exosome biogenesis.

The YBX1 protein has three distinct domains, one of which, the cold-shock domain (CSD) appears to be the principal site for RNA binding, including at least one critical residue, F85, required for binding miR223 as well as other RNAs (*Lyons et al., 2016*). The C-terminal domain (CTD) includes an intrinsically disordered domain (IDR) that promotes the formation of a liquid-liquid phase separation likely responsible for the organization of YBX1 in P-bodies (*Liu et al., 2021*). This domain does not itself interact with RNA, but it appears to facilitate the folding or stabilization of the CSD to promote high affinity binding to miR223.

In other work using a similar approach, we identified two separate sorting signals, a 5'UGGA and a 3'UUU, on miR122 to which the RNA-binding protein La binds *en route* to secretion in exosomes by the breast cancer cell line MDA-MB-231 (*Temoche-Diaz et al., 2019*). Other distinct sorting signals and their cognate RNA-binding proteins have been documented in different cell lines. miRNAs with a GGAG sorting motif recognized by a sumolyated form of hnRNPA2B1 was shown to be enriched in exosomes (*Villarroya-Beltri et al., 2013*). Another sequence, AAUGC, was found to be enriched in

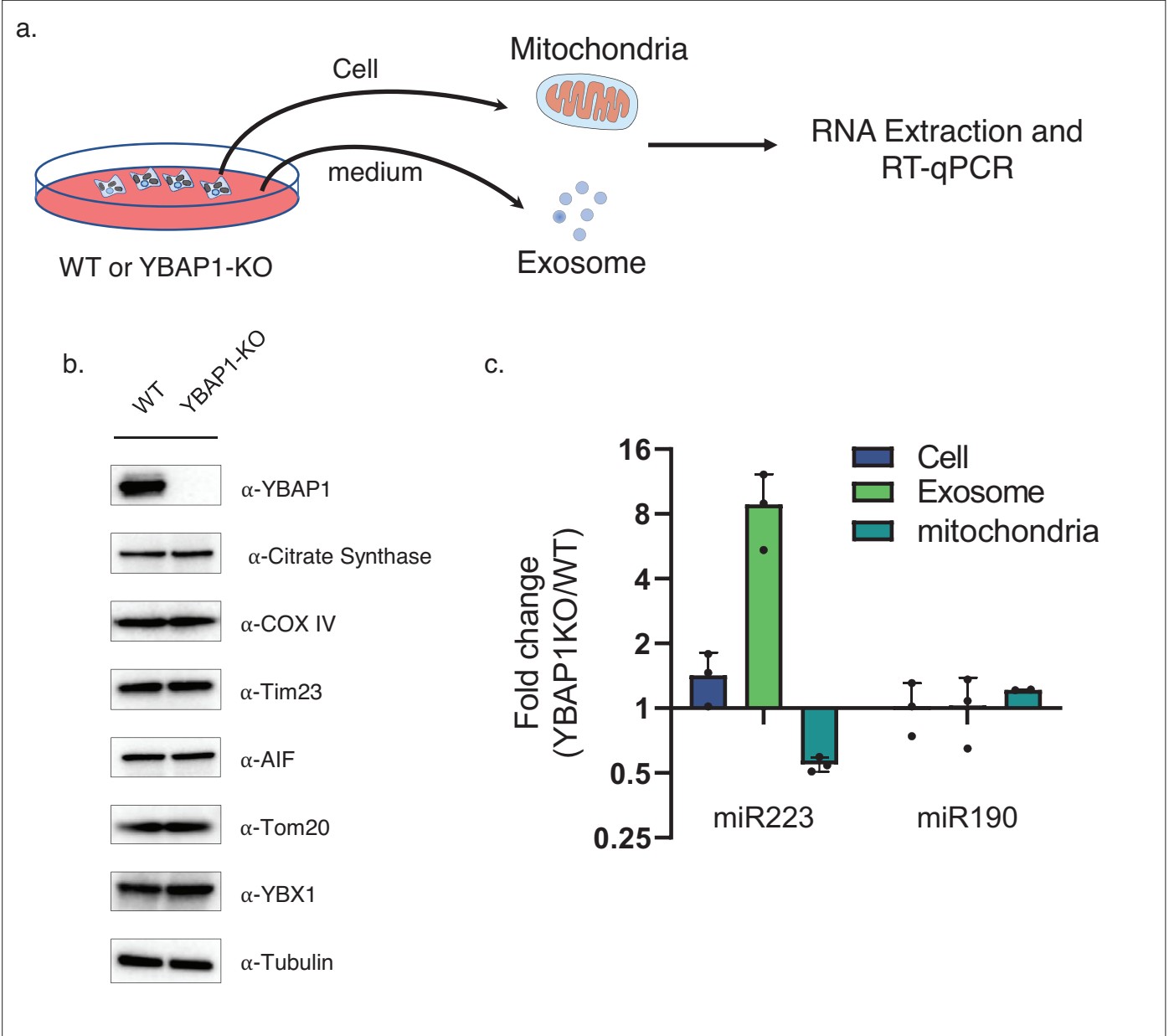

**Figure 5.** YBAP1 sequesters miR223 which is released and secreted in YBAP1 KO cells. (**a**) Schematic shows exosome and mitochondria purification from 293 T WT cells and YBAP1 knock out cells for RT-qPCR analysis. (**b**) Analysis of 293 T WT and CRISPR/Cas9 genome-edited cells by immunoblot for YBAP1, YBX1 and mitochondrial markers (**c**) RT-qPCR analysis of miR223 enrichment in mitochondria purified from 293 T WT cells and YBAP1 KO cells relative to cell lysate. Data are plotted from three independent experiments and error bars represent standard deviations. (**d**) RT-qPCR analysis of miR223 and miR190 fold change in cells, purified mitochondria and exosomes from 293 T WT cells and YBAP1 KO cells. Data are plotted from three independent experiments and error bars represent standard deviations.

The online version of this article includes the following source data and figure supplement(s) for figure 5:

**Source data 1.** Uncropped immunoblot images corresponding to *Figure 5*.

**Figure supplement 1.** Knockout of YBAP1 did not change exosome secretion.

exosomal miRNA and dependent on the RNA-binding protein FMR1 for miRNA secretion (*Wozniak et al., 2020*). Diverse cell lines and likely tissues appear to invoke distinct sorting signals decoded by different RNA-binding proteins. Many of the proteins may engage in biomolecular condensates such as P-bodies as a mechanism to sort RNAs for secretion (*Liu et al., 2021*).

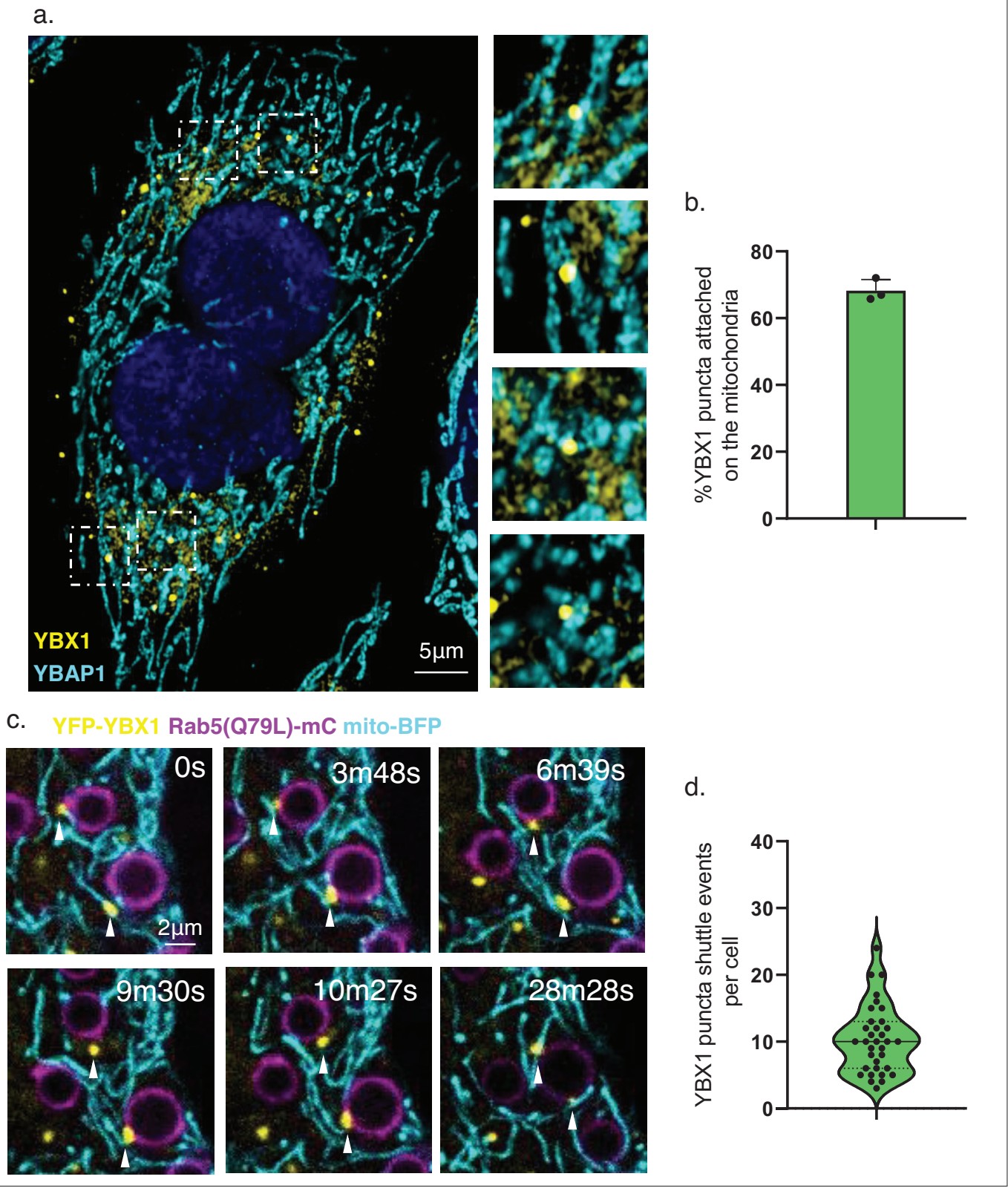

**Figure 6.** YBX1 puncta relocalize from mitochondria to endosomes. (**a**) YBX1 puncta on the mitochondria. U2OS cells were stained with anti-YBX1 and anti-YBAP1 antibodies and observed by confocal microscopy. The right panel shows enlarged regions of interest from the left panel. Scale bar, 5 μm. (**b**) The statistics are of the percentage of YBX1 puncta detected in proximity to mitochondria. N=30 cells. (**c**) YBX1 puncta relocalize from mitochondria to endosomes. U2OS cells overexpressed YFP-YBX1, Rab5(Q79L)-mCherry and mito-BFP. Time-lapse images were acquired with a Zeiss LSM900 confocal microscope. Scale bar, 2 μm. (**d**) The statistics are of YBX1 puncta shuttle events per cell. The data was represented as violin plots. N=34 cells.

miR223 appears to be an example of a number of small nuclear-encoded RNAs localized to mitochondria (*Jeandard et al., 2019*). In some cases these RNAs, such as tRNAs, serve an essential function such as in mitochondrial protein synthesis, however, for miRNAs with no obvious mitochondrial genome target, the function remains unclear (*Jeandard et al., 2019*). Nonetheless, others have documented the localization of these miRNAs enclosed within the mitochondrion and in the case of miR223, it appears to be tightly associated with the inner membrane. The exact organization and function of miR223 in this location remains to be investigated but in the context of exosomal secretion, the mitochondrial localization appears to serve as a reservoir.

In relation to the mitochondrial localization of miR223, we found a mitochondrial RNA-binding protein, YBAP1, that copurified with a tagged form of YBX1 expressed in HEK293 cells. YBAP1 has previously been reported to interact with YBX1 and independently found associated with mitochondria where its localization is dependent on an N-terminal transit peptide sequence (*Muta et al., 1997*). The association of mitochondrial YBAP1 and cytoplasmic YBX1 was reproduced by coexpression of recombinant forms of the two proteins in baculovirus-infected SF9 cells (*Figure 4—figure supplement 1b*). YBAP1 binds miR223 selectively but with an affinity significantly below that of YBX1 (*Figure 4h–j*). Although YBX1 does not localize to the mitochondrion, the stable interaction of the complex may suggest a transient relationship, perhaps during the biogenesis of YBAP1 as it transits from the cytoplasm into the mitochondrion.

A functional relationship between YBAP1 and YBX1 is suggested by the reduction in miR223 in mitochondria and increase in secretion of miR223 in exosomes secreted from YBAP1 KO cells (*Figure 5*). In contrast, removal of mitochondria by treatment of cells with CCCP resulted in an increase in cytoplasmic miR223 at the expense of secretion in exosomes (*Figure 3*). Although YBAP1 may facilitate the retention of miR223 within mitochondria, mitochondrial RNA import and export may serve an

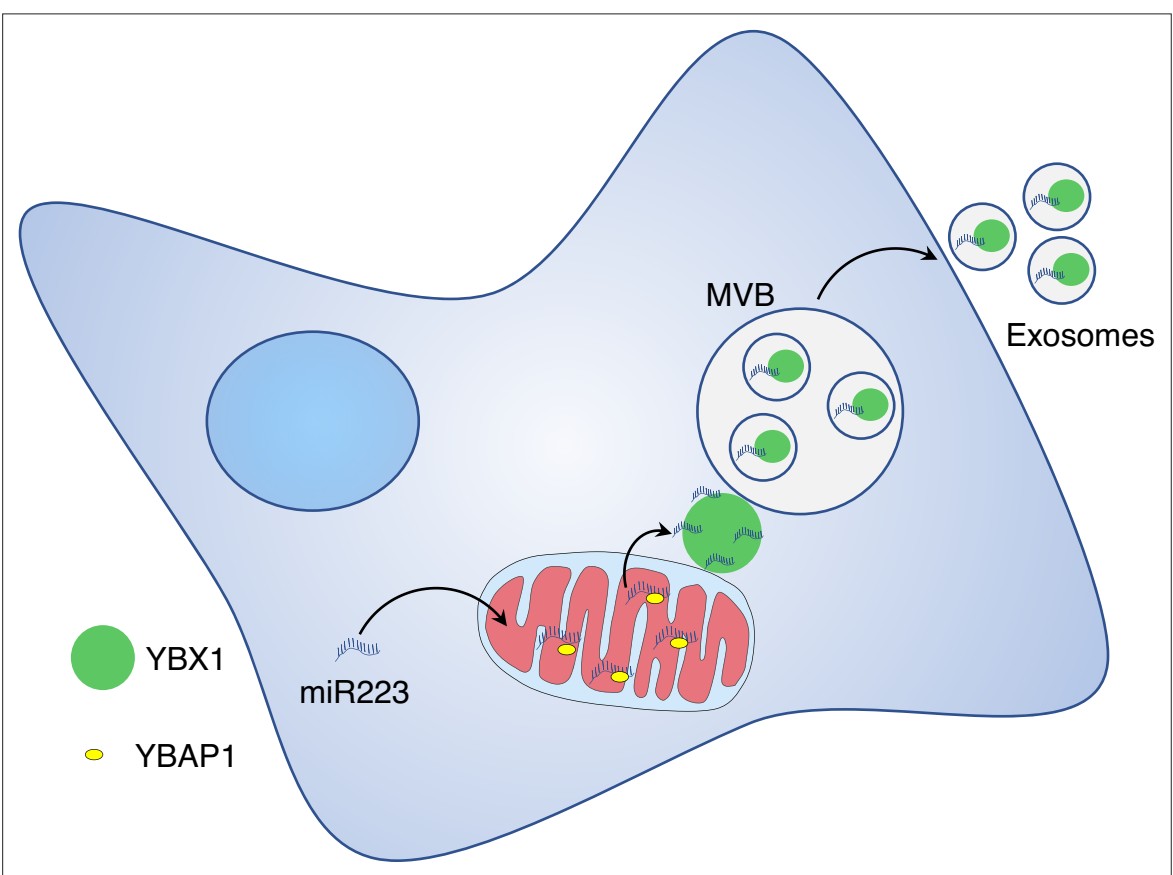

**Figure 7.** Diagram representing a model of miR223 sorting from mitochondria into exosomes. Stages in the transfer of miR223 from mitochondria. Cytosolic miR223 is enriched in mitochondria where it may be sequestered by a weak interaction with YBAP1. Cytoplasmic YBX1 interacts more tightly with miR223 which may drive the removal of miR223 from mitochondria. YBX1 in RNA granules may accumulate miR223 removed from mitochondria. YBX1 puncta may give rise to small particles carrying miR223 for uptake into endosomes and secretion in exosomes.

independent role in the selective capture of miRNAs by YBX1 in cytoplasmic P-body condensates. YBX1 puncta appear to shuttle between mitochondria and endosomes at which point miR223 bound to YBX1 may be further sorted into invaginations budding into the interior of endosomes.

The highly selective nature of miRNA sorting and secretion in exosomes suggests an important role in the trafficking of miRNAs between cells. Numerous studies have suggested a role for secreted miRNAs in recipient cells (*Cha et al., 2015*; *Mittelbrunn et al., 2011*; *Pegtel et al., 2010*; *Valadi et al., 2007*). Nonetheless, as miRNAs ordinarily act stoichiometrically on target mRNAs, the extremely low abundance and copy number of miRNAs/vesicle is hard to reconcile with such a functional role of secreted miRNA (*Chevillet et al., 2014*; *Shurtleff et al., 2017*). Our observation that the bulk of cellular miR223 is held within mitochondria suggests an alternative role in some structural or regulatory process, perhaps essential for mitochondrial homeostasis, controlled by the selective extraction of unwanted miRNA into RNA granules and further by secretion in exosomes (*Figure 7*).

# Materials and methods

## Key resources table

| Reagent type (species) or resource | Designation | Source or reference | Identifiers | Additional information |
|---|---|---|---|---|
| Cell line (Spodoptera frugiperda) | Sf9 | Other | | Cell culture facility at UC Berkeley |
| Cell line (*Homo sapiens*) | HEK 293T cells | Other | | Cell culture facility at UC Berkeley |
| Cell line (*Homo sapiens*) | HEK 293T-YBX1 KO cells | Other | | Obtained by CRISPR-Cas9 in Schekman Lab |
| Cell line (*Homo sapiens*) | HEK 293T-YBAP1 KO | This study | | Obtained by CRISPR-Cas9 in Schekman Lab |
| Cell line (*Homo sapiens*) | HEK 293T-3xHA-EGFP-OMP25 | This study | | Obtained by overexpression of pLJM1-3XHA-EGFP-OMP25 in Schekman lab |
| Cell line (*Homo sapiens*) | U-2OS cells | Other | | Cell culture facility at UC Berkeley |
| Cell line (*Homo sapiens*) | U-2OS Parkin-GFP cells | This study | | Obtained by overexpression of Parkin-GFP in Schekman lab |
| Recombinant DNA reagent | pFastBac His6 MBP N10 TEV LIC cloning vector (4 C) | Addgene | RRID: Addgene_30116 | N/A |
| Recombinant DNA reagent | Tom22-mCherry (plasmid) | This study | | Gift of Dr Li Yu lab |
| Recombinant DNA reagent | His-MBP-YBX1 (plasmid) | This study | | To express YBX1 in insect cells. Plasmid maintained in Schekman lab |
| Recombinant DNA reagent | His-MBP-YBX1(1–51) (plasmid) | This study | | To express YBX1(1–51) in insect cells. Plasmid maintained in Schekman lab |
| Recombinant DNA reagent | His-MBP-YBX1(52–129) (plasmid) | This study | | To express YBX1(52–129) in insect cells. Plasmid maintained in Schekman lab |
| Recombinant DNA reagent | His-MBP-YBX1(130–324) (plasmid) | This study | | To express YBX1(1–51) in insect cells. Plasmid maintained in Schekman lab |
| Recombinant DNA reagent | His-MBP-YBX1(1–129) (plasmid) | This study | | To express YBX1(1–129) in insect cells. Plasmid maintained in Schekman lab |
| Recombinant DNA reagent | His-MBP-YBX1(F85A) (plasmid) | This study | | To express YBX1(F85A) in insect cells. Plasmid maintained in Schekman lab |
| Recombinant DNA reagent | His-MBP-YBAP1 (plasmid) | This study | | To express YBAP1 in insect cells. Plasmid maintained in Schekman lab |
| Recombinant DNA reagent | Mito-BFP | This study | | Gift of Dr. Samantha Lewis lab |
| Recombinant DNA reagent | mCherry-Rab5(Q79L) (plasmid) | Addgene | RRID: Addgene_35138 | |
| Recombinant DNA reagent | pLJM1-3XHA-EGFP-OMP25 | This study | | To express 3xHA-EGFP-OMP25 in HEK293T cells. Plasmid maintained in Schekman lab |

*Continued on next page*

*Continued*

| Reagent type (species) or resource | Designation | Source or reference | Identifiers | Additional information |
|---|---|---|---|---|
| Antibody | Anti-YBX1 (Rabbit polyclonal) | Abcam | RRID: AB_1950384 | WB 1:1000 |
| Antibody | Anti-YBAP1 (Mouse monoclonal) | Santa Cruz | RRID: AB_10611471 | WB 1:1000 |
| Antibody | Anti-YBAP1(Rabbit polyclonal) | Thermo Fisher Scientific | RRID: AB_2638956 | WB 1:1000 |
| Antibody | Anti-Tim23 (Mouse monoclonal) | BD Biosciences | RRID: AB_398754 | WB 1:1000 |
| Antibody | Anti-Tom20 (Mouse monoclonal) | Abcam | RRID: AB_945896 | WB 1:1000 |
| Antibody | Anti-Calnexin (Rabbit polyclonal) | Abcam | RRID: AB_2069006 | WB 1:2000 |
| Antibody | Anti-HA (Rabbit monoclonal) | Cell Signaling | RRID: AB_1549585 | WB 1:1000 |
| Antibody | Anti-COX IV (Rabbit Monoclonal) | Cell signaling | RRID: AB_2085424 | WB 1:1000 |
| Antibody | Anti-Citrate Synthase (Rabbit monoclonal) | Cell signaling | RRID: AB_2665545 | WB 1:1000 |
| Antibody | Anti-Rab5 (Rabbit monoclonal) | Cell signaling | RRID: AB_2300649 | WB 1:1000 |
| Antibody | Anti-LAMP1 (Rabbit monoclonal) | Cell signaling | RRID: AB_2687579 | WB 1:1000 |
| Antibody | Anti-GRP78 (Rabbit polyclonal) | Abcam | RRID: AB_2119834 | WB 1:3000 |
| Antibody | Anti-GADPH (Rabbit monoclonal) | Cell signaling | RRID: AB_561053 | WB 1:5000 |
| Antibody | Anti-alpha Tubulin (Mouse monoclonal) | Abcam | RRID: AB_2241126 | WB 1:5000 |
| Antibody | Anti-beta Actin (Mouse monoclonal) | Abcam | RRID: AB_449644 | WB 1:5000 |
| Software, algorithm | Image Studio Lite | LICOR | | https://www.licor.com/bio/image-studio-lite/ |
| Software, algorithm | FIJI | NIH | RRID: SCR_002285 | https://fiji.sc/ |
| Software, algorithm | Prism 9 | GraphPad | RRID: SCR_002798 | https://www.graphpad.com |

## Cell lines and cell culture

All immortalized cell lines were obtained from the UC-Berkeley Cell Culture Facility and were confirmed by short tandem repeat (STR) profiling and tested negative for mycoplasma contamination. HEK 293T cells were cultured in DMEM with 10% FBS(VWR), NEAA (Gibco, Cat No: 11140050) and 1 mM Sodium Pyruvate (Gibco, Cat No: 11360070). For exosome production, we seeded cells at 10~20% confluency in 150 mm tissue culture dishes (Fisher Scientific, Cat No: 12-565-100) containing 30 ml of exosome-free medium. Exosomes were collected from 80% confluent cells (~48 hr).

## Exosome purification

Conditioned medium was harvested from 80% to 90% confluent HEK 293T cultured cells. All procedures were performed at 4 °C. Cells and large debris were removed by centrifugation in a Sorvall R6 +centrifuge at 1000x*g* for 15 min followed by 10,000x*g* for 15 min using a FIBERlite F14−6x500 y rotor. The supernatant fraction was then centrifuged onto a 60% sucrose cushion in a buffer with 10 mM HEPES (pH 7.4) and 0.85% w/v NaCl at ~100,000 x *g* (28,000 RPM) for 1.5 h in a SW32Ti rotor. The interface over the sucrose cushion was collected and pooled for an additional centrifugation onto a 2 ml 60% sucrose cushion at ~120,000 x *g* (31,500 RPM) for 15 h using an SW41Ti rotor. The first collected interface was measured by refractometry and adjusted a sucrose concentration not exceeding 21%. For bulk purification, the EVs collected from the interface over the sucrose cushion after the first SW41Ti centrifugation were mixed with 60% sucrose to a final volume of 10 ml (the concentration of sucrose ~50%). One ml of 40% and 1 ml of 10% sucrose

were sequentially overlaid and the samples were centrifuged at ~150,000 x *g* (36,500 rpm) for 15 h in an SW41Ti rotor. The exosomes were located at the 10%/40% interface and collected for RNA extraction or immunoblot.

## Density gradient isolation of mitochondria and mitoplasts

Mitochondria were isolated according to a well-established published protocol (*Wang et al., 2020*). HEK293T Cells were harvested at 80% confluency and were homogenized in 6 vol of HB buffer (225 mM mannitol, 25 mM sucrose, 0.5% BSA, 0.5 mM EGTA, 30 mM Tris–HCl, pH 7.4, and protease inhibitors) in a prechilled Dounce homogenizer (Kontes). The lysate was centrifuged and the postnuclear supernatant was collected. Crude mitochondria were centrifuged at 6300 x *g* for 8 min, washed once with MRB buffer (250 mM mannitol, 0.5 mM EGTA, and 5 mM HEPES, pH 7.4), resuspended in 1 ml MRB buffer, laid over a 30% Percoll solution (9 ml) and centrifuged at 95,000 g for 45 min. The buoyant, purified fraction of mitochondria was collected for further analysis. For mitoplast purification, crude mitochondria were resuspended into 10 vol MRB buffer with 0.2 mg/ml digitonin and incubated on ice for 15 min. Digitonin-treated crude mitochondria were laid over a 30% Percoll solution (9 ml) and centrifuged at 95,000 *g* for 45 min. A buoyant, purified fraction of mitoplasts was collected for further analysis.

## YBAP1 immunoprecipitation from mitochondria

After the Percoll gradient purification, the enriched mitochondria were diluted 2 x into MRB buffer and centrifuged at 12,000 *g* for 10 min. The mitochondrial pellet was lysed in 0.5 ml RIPA buffer (50 mM Tris-HCl, pH 7.5, 150 mM NaCl, 2 mM EDTA, 1% DDM, 1 mM PMSF) containing protease inhibitors (1 mM 4-aminobenzamidine dihydrochloride, 1 µg/ml antipain dihydrochloride, 1 µg/ml aprotinin, 1 µg/ml leupeptin, 1 µg/ml chymostatin, 1 mM phenylmethylsulphonyl fluoride, 50 µM N-tosyl-L-phenylalanine chloromethyl ketone and 1 µg/ml pepstatin) and RNase inhibitors, followed by centrifugation at 12,000 *g* for 10 min. Supernatant fractions were incubated with 10 µl washed protein A Dynabeads (ThermoFisher Scientific, Catalog number: 10001D) and 0.5 µg mouse monoclonal IgG antibody and rotated at 4 °C for 1 h. A magnetic rack was used to remove protein A beads and the resulting supernatant fractions were incubated with 40 µl washed protein A Dynabeads and 4 µg YBAP1 antibody or mouse IgG antibody and rotated at 4 °C overnight. The beads were collected using a magnetic rack, washed 3 x with 1 ml of RIPA buffer, and collected for immunoblot and RNA extraction.

## Mitochondria immunoprecipitation

Mito-IP was performed as previously described with slight modifications (*Chen et al., 2016*). The mito-IP cell-line was grown to ~90% confluency in 15 cm dishes. All the subsequent steps of mito-IP were performed using ice-cold buffers either in a cold-room or on ice. Cells ($2 \times 10^7$) were washed twice with 10 ml of PBS and then harvested in 10 ml of mito-IP buffer (10 mM KH2PO4, 137 mM KCl) containing protease inhibitors (1 mM 4-aminobenzamidine dihydrochloride, 1 µg/ml antipain dihydrochloride, 1 µg/ml aprotinin, 1 µg/ml leupeptin, 1 µg/ml chymostatin, 1 mM phenylmethyl-sulphonyl fluoride, 50 µM N-tosyl-L-phenylalanine chloromethyl ketone and 1 µg/ml pepstatin) and TCEP 0.5 mM. The final mito IP buffer also contained 6 ml of OptiPrep (Sigma) per 100 ml. Cells were collected at 700x*g* for 5 min and resuspended in 1 ml of mito-IP buffer per 15 cm plate and then lysed using 5–10 passes through a 22 G needle. A post-nuclear supernatant (PNS) fraction was obtained after centrifuging the lysate at 1500xg for 10 min to remove unbroken cells and nuclei. Whenever necessary, a fraction of PNS was saved for immunoblot analysis. The resulting PNS was incubated with 100 µl of anti-HA magnetic beads (Sigma) pre-equilibrated in the mito-IP buffer in 1.5 ml microcentrifuge tubes and then gently rotated on a mixer for 15 min. The beads were collected using a magnetic rack and washed 3 x for 5 min with 1 ml of mito-IP buffer.

For mitoplast purification by osmotic shock, the supernatant was discarded after the final wash of the mito-IP sample, and the beads were gently resuspended in 200 µl of hypotonic osmotic shock buffer (OSB) containing 20 mM HEPES at pH 7.4. The resuspended sample was incubated on ice for 30 min and then the beads were centrifuged at 15,000 *g* for 15 min to sediment mitochondria/ mitoplasts. Beads were then resuspended in 100 µl of KPBS and proteinase K was added to achieve a final concentration of 10 µg/ml and samples were incubated on a rotating mixer at 4 °C for 15 min.

Subsequently, PMSF was added to a final concentration of 1 mM, along with a protease inhibitor cocktail, and the sample was incubated on ice for 5 min. Next, 2.5 units of RNase ONE (Promega) was added to the sample, which was further incubated on a rotating mixer at room temperature for 15 min. For protein analysis, the sample was eluted directly in the SDS loading buffer. Alternatively, Trizol was added to the sample to stop the reaction for RNA purification.

To assess their quality, we assayed the immunoprecipitated mitochondria for the following protein markers by immunoblotting using the rabbit primary antibodies- anti-HA (1:1000), anti-COX IV (1:1000), anti-TOM20 (1:1000), anti-Citrate Synthase (1:1000), anti-RAB5 (1:1000), anti-LAMP1 (1:2000), anti-GAPDH (1:5000), and anti-GRP78 (Abcam) (1:3000). All the above antibodies were sourced from Cell Signaling Technology, unless stated otherwise. As a negative control, non-transduced HEK293T cells were used in these experiments to assess the non-specific capture of the marker proteins.

### Mitochondrial fractionation

One mito-IP was performed per sample as described above. To ensure an even distribution of mitochondria across the samples, we pooled washed beads from all the IPs and equally distributed aliquots for subsequent treatments. Mitochondria were lysed in a 50 µl final volume using either 1% vol/vol Triton X-100 (final concentration) or by three sequential rounds of freeze/thaw using liquid-nitrogen, as indicated. Urea was added to a final concentration of 3 M. After a 10 min incubation on ice, samples were centrifuged at 15000xg for 15 min and supernatant and pellet fractions were collected as indicated. The total fractionated mitochondria were analyzed by immunoblotting for various mitochondrial markers. To analyze the specific RNA content of total or fractionated mitochondria, we extracted RNA using Trizol (Invitrogen) as per manufacturer's recommendations followed by q-PCR.

### In vitro packaging of miR223 and miR223mut
#### Preparation of membranes and cytosol

The membrane and cytosol fractions were prepared from HEK293T cells as previously described with slight modifications (*Shurtleff et al., 2016*; *Temoche-Diaz et al., 2020*). All steps were carried out in either the cold-room or on ice using ice-cold buffers and pre-chilled equipment. Briefly, HEK293T cells (80% confluency) were washed twice with PBS and then harvested in the homogenization buffer (HB) (250 mM sorbitol, 20 mM HEPES-KOH pH 7.4) containing protease inhibitor cocktail (1 mM 4-aminobenzamidine dihydrochloride, 1 µg/ml antipain dihydrochloride, 1 µg/ml aprotinin, 1 µg/ml leupeptin, 1 µg/ml chymostatin, 1 mM phenylmethylsulphonyl fluoride, 50 µM N-tosyl-L-phenylalanine chloromethyl ketone and 1 µg/ml pepstatin). The cell pellet was obtained by centrifuging the cells at 500xg for 5 min. After discarding the supernatant, cells were weighed and resuspended in two volumes of HB followed by lysis with 5–10 passages through a 22 G needle. The lysate was centrifuged at 1500xg for 10 min to remove unbroken cells and nuclei to obtain a PNS which was then centrifuged at 20,000xg for 30 min to obtain a membrane fraction. The supernatant from above was centrifuged at 150,000xg for 30 min using a TLA-55 rotor (Beckman Coulter ) and the resulting supernatant was used as the cytosol fraction (~6 mg protein/ml). Membranes from the first 20,000xg sedimentation were resuspended in 1 ml of HB and centrifuged again at 20,000xg for 30 min. The pellet fraction was resuspended in one volume of HB and rested on an ice block for a minimum of 10 min until the insoluble components and debris settled at the bottom of the tube. The finely resuspended material in the resulting supernatant fraction was then transferred to a new microcentrifuge tube (to avoid the settled debris) and was used as the membrane fraction.

### Preparation of radiolabeled miR223 and miR223mut substrates

HPLC purified miR223 and miR223mut oligos were obtained from IDT. A stock solution of these oligos (1 µl of a 10 µM) was 5'-end-labeled using T4PNK (NEB) and 5 µl of ATP, [γ–32P]- 6000 Ci/mmol 10mCi/ml EasyTide (PerkinElmer BLU502Z250UC) as per manufacturer's recommendations in a 50 µl reaction volume. T4PNK was heat-inactivated at 70 °C for 15 min. Unincorporated radionucleotides were removed by passing through PerformaTM spin columns (EdgeBio). The flow-through (radiolabeled substrate) was collected and stored at –20 °C until further use.

## In vitro miR223 packaging reaction

Wherever indicated, 10 µl of cytosol (~5 mg/ml), 17 µl of membranes, 2 µl of radiolabeled substrate, 9 µl of 5 x incorporation buffer (400 mM KCl, 100 mM CaCl$_2$, 60 mM HEPES-NaOH, pH 7.4, 6 mM, MgOAc), 4.5 µl of 10 x ATP regeneration system (400 mM creatine phosphate, 2 mg/ml creatine phosphokinase, 10 mM ATP, 20 mM HEPES pH 7.2, 250 mM sorbitol, 150 mM KOAc, 5 mM MgOAc), 1 µl of ATP (100 mM, Promega), 0.5 µl of GTP (100 mM, Promega), 1 µl of Ribolock (40 U/µl, Invitrogen) were mixed to setup a 45 µl in vitro packaging reaction. In samples without the cytosol or membranes, the final reaction volumes were adjusted to 45 µl using HB. The reactions were incubated at either 30 °C or on ice, as indicated, for 15 min. Following the incubation, the indicated samples were subjected to RNAse ONE(Promega) using 10 U of the enzyme in the presence of urea (300 mM final concentration) in a total reaction volume of 60 µl. Wherever indicated, TritonX-100 was added to a final concentration of 1%. The RNAse treatments were carried out for 20 min at 30 °C followed by RNA extraction using DirectZol (Zymo Research) kits as per manufacturer's protocol. RNA was precipitated overnight at –20 °C by the addition of 3 vol of ethanol, 1/10th volume of 3 M sodium acetate (pH5.2) and 30 µg Glycoblue reagent (Invitrogen). Precipitated RNA was sedimented at 16,000xg for 30 min followed by washing with ice-cold 70% ethanol. The RNA pellet was resuspended in 2 X RNA loading dye (NEB) and heated for 5 min at 70 °C. RNA was separated using a 15% denaturing polyacrylamide gel, followed by gel drying using a vacuum gel dryer (Model 583, Biorad). Radioactive bands were visualized by phosphorimaging using a Kodak storage phosphor screen and the Pharos FX Plus Molecular Imager (Biorad).

## Immunoblots

Cell lysates and other samples were prepared by adding 2% SDS and heated at 95 °C for 10 min. Protein was quantified using a BCA Protein Assay Kit (Thermo Fisher Scientific) and appropriate amounts were mixed with 5 x SDS loading buffer. Samples were heated at 95°C for 10 min and separated on 4–20% acrylamide Tris-glycine gradient gels (Life Technologies). Proteins were transferred to PVDF membranes (EMD Millipore, Darmstadt, Germany) and the membrane was blocked with 5% fat-free milk powder in TBST and incubated for 1 h at room temperature or overnight at 4 °C with primary antibodies. Blots were then washed in three washes of TBST for 10 min each. Membranes were incubated with anti-rabbit or anti-mouse secondary antibodies (GE Healthcare Life Sciences, Pittsburgh, PA) for 1 hr at room temperature and rinsed in three washes of TBST for 10 min each. Blots were developed with ECL-2 reagent (Thermo Fisher Scientific). Primary antibodies used in this study were as follows: anti-Tim23 (BD, 611222), Calnexin (Abcam, ab22595), Actin (Abcam, ab8224), Tubulin (Abcam, ab7291), YBAP1 (Santa Cruz Biotechnology, sc-271200).

## Immunofluorescence

Cells were cultured on 12 mm round coverslips (corning) and were fixed with 4% EM-grade paraformaldehyde (Electron Microscopy Science, Hatfield, PA) in PBS pH7.4 for 10 min at room temperature. Cells were then washed 3 x with PBS for 10 min each, treated with permeabilizing buffer (10% FBS in PBS) containing 0.1% saponin for 20 min and treated in blocking buffer for 30 min. Subsequently, cells were incubated with primary antibodies in permeabilizing buffer for 1 hr at room temperature, washed 3 x with PBS for 10 min each and incubated with secondary antibodies in permeabilizing buffer for 1 hr at room temperature and finally washed 3 x with PBS for 10 min each. Cells were mounted on slides with Prolong Gold with DAPI (Thermo Fisher Scientific, P36931). Primary antibodies used in the immunofluorescence studies were as follows: anti-YBX1 (Abcam, ab12148), YBAP1 (Santa Cruz Biotechnology, sc-271200). Images were acquired with Zeiss LSM900 confocal microscope and analyzed with the Fiji software (http://fiji.sc/Fiji).

## Quantitative real-time PCR

Cellular and EV RNAs were extracted using a mirVana miRNA isolation kit (Thermo Fisher Scientific, AM1560) or Direct-zol RNA Miniprep kits (Zymo Research). Taqman miRNA assays for miRNA detection were purchased from Life Technologies. Assay numbers were: hsa-miR-223–3 p, 002295; hsa-mir-190–5 p, 000489; U6 snRNA, 001973. Total RNAs were quantified using RNA bioanalyzer (Agilent). Taqman qPCR master mix with no Amperase UNG was obtained from Life Technologies for reverse transcription. For mRNA, RevertAid First Strand cDNA Synthesis Kit (Thermo Scientific, K1621)

was used for reverse transcription. COX1 qPCR primers: Forward-5'-TCTCAGGCTACACCCTAGAC CA-3', Reverse-5'-ATCGGGGTAGTCCGAGTAACGT-3'. GAPDH qPCR primers: Forward-5'-CTGA CTTCAACAGCGACACC-3', Reverse-5'-TAGCCAAATTCGTTGTCATACC-3'. Quantitative real-time PCR was performed using an QuantStudio 5 Real-Time PCR System (Applied Biosystems).

## Protein purification

Twin Strep tag hybrid YBX1 was expressed and the protein was isolated 48 hr after PEI-mediated transfection of 293T cells. Cells were resuspended in PBS and collected by centrifugation for 5 min at 600 *g*. Pellet fractions were resuspended in 35 ml lysis buffer (50 mM Tris-HCl (pH 8),150mM NaCl,1mM EDTA, 2 mM DTT, 1 mM PMSF and 1 x protease inhibitor cocktail). After sonication of the cell suspension the crude lysate was centrifuged for 60 min at 20,000 rpm at 4 °C. The resulting supernatant fraction was incubated with 2 ml Strep-Tactin Sepharose resin (IBA, 2-1201-010) for 1 h. Strep-Tactin Sepharose resin samples were transferred to columns (18 ml) and protein-bound beads were washed with 60 ml wash buffer (50 mM Tris-HCl (pH 8), 500 mM NaCl, 1 mM EDTA, 2 mM DTT) until no protein was eluted as monitored by the Bio-Rad protein assay (Bio-Rad, Catalog #5000006). Proteins were eluted with 10 ml elution buffer (50 mM Tris-HCl (PH = 8),150mM NaCl, 10 mM desthi-obiotin, 1 mM EDTA, 2 mM DTT) and concentrated using an Amicon Ultra Centrifugal Filter Unit (50 kDa, 4 ml) (Fisher Scientific, EMD Millipore). Proteins were further purified by gel filtration chroma-tography (Superdex-200, GE Healthcare) with columns equilibrated in storage buffer (50 mM Tris-HCl 7.4, 500 mM KCl, 5% glycerol, 1 mM DTT). Peak fractions corresponding to the appropriate fusion protein were pooled, concentrated, and distributed in 10 μl aliquots in PCR tubes, flash-frozen in liquid nitrogen and stored at –80 °C. Protein concentration was determined by known concentrations of BSA assessed by Coomassie Blue staining.

Tagged (6xHis) and maltose-binding protein hybrid genes were expressed in baculovirus-infected SF9 insect cells (*Lemaitre et al., 2019*). Insect cell cultures (1 l, 1x10⁶ cells/ml) were harvested 48 h after viral infection and collected by centrifugation for 20 min at 2000 rpm. The pellet fractions were resuspended in 35 ml lysis buffer (50 mM Tris-HCl 7.4, 0.5 M KCl, 5% glycerol, 10 mM imidazole, 0.5 μl/ml Benzonase nuclease (Sigma, 70746–3), 1 mM DTT, 1 mM PMSF and 1 x protease inhibitor cocktail). Cells were lysed by sonication and the crude lysate was centrifuged for 60 min at 20,000 rpm at 4 °C. After centrifugation, the supernatant fraction was incubated with 2 ml Ni-NTA His-Pur resin (Thermo Fisher, PI88222) for 1 hr. Ni-NTA resin samples were transferred to columns (18 ml) and protein-bound beads were washed with 60 ml lysis buffer until no protein was eluted as monitored by the Bio-Rad protein assay (Bio-Rad, Catalog #5000006). Proteins were eluted with 10 ml elution buffer (50 mM Tris-HCl 7.4, 0.5 M KCl, 5% glycerol, 500 mM imidazole). The eluted sample was incubated with 2 ml amylose resin (New England Biolabs, E8021L) for 1 hr at 4 °C. Amylose resin samples were transferred to columns and protein-bound beads were washed with 60 ml lysis buffer until no protein was eluted as monitored by the Bio-Rad protein assay. Proteins were eluted with 10 ml elution buffer (50 mM Tris-HCl 7.4, 500 mM KCl, 5% glycerol, 50 mM maltose) and were concentrated using an Amicon Ultra Centrifugal Filter Unit (50 kDa, 4 ml) (Thermo Fisher Scientific, EMD Millipore). Proteins were further purified by gel filtration chromatography (Superdex-200, GE Healthcare) with columns equilibrated in storage buffer (50 mM Tris-HCl 7.4, 500 mM KCl, 5% glycerol, 1 mM DTT). Peak fractions corresponding to the appropriate fusion protein were pooled, concentrated, and distributed in 10 μl aliquots in PCR tubes, flash-frozen in liquid nitrogen and stored at –80 °C. Protein concentration was determined by known concentrations of BSA based on Coomassie blue staining.

## CRISPR/Cas9 genome editing

A pX330-based plasmid expressing Venus fluorescent protein (*Shurtleff et al., 2016*) was used to clone the gRNAs targeting YBAP1. A CRISPR guide RNA targeting the first exon of the YBAP1 open reading frame was designed following the CRISPR design website (http://crispor.tefor.net/crispor.py): CGCTGCGTGCCCCGTGTGCT. Oligonucleotides encoding gRNAs were annealed and cloned into pX330-Venus as described (*Cong et al., 2013*). HEK293T cells were transfected by Lipofectamine 2000 for 48 hr at low passage number, trypsinized and sorted for single, Venus positive cells in 96-well plates by a BD Influx cell sorter. YBAP1 knockout candidates were confirmed by immunoblot. HEK 293T YBX1 knockout cells were generously provided by Dr. Xiaoman Liu (*Liu et al., 2021*).

## Electrophoretic mobility shift assay

Fluorescently labeled RNAs (5'-IRD800CWN) for detecting free and protein-bound RNA were ordered from Integrated DNA Technologies (IDT, Coralville, IA). EMSA was performed as described with some modification (*Rio, 2014*). Briefly, 1 nM of IRD800CWN-labeled RNA was incubated with increasing amounts of purified proteins, ranging from 500 pM - 1 µM. Buffer E was used in this incubation (25 mM Tris pH8.0, 100 mM KCl, 1.5 mM MgCl2, 0.2 mM EGTA, 0.05% Nonidet P-40, 1 mM DTT, 5% glycerol, 50 µg/ml heparin). Reactions were incubated at 30 °C for 30 min then chilled on ice for 10 min. Samples were mixed with 6 x loading buffer (60 mM KCl, 10 mM Tris pH 7,6, 50% glycerol, 0.03% (w/v) xylene cyanol). Mixtures (5 µl) were loaded onto a 6% native polyacrylamide gel and electrophoresed at 200 V for 45 min in a cold room. The fluorescence signal was detected using an Odyssey CLx Imaging System (LI-COR Biosciences, Lincoln, NE). The software of the Odyssey CLx Imaging System was used to quantify fluorescence. To calculate Kds, we fitted used Hill equations with quantified data points.

## CD63-Nluc exosome secretion assay

The CD63-Nluc exosome secretion assay was carried out as described (*Williams et al., 2023*). Briefly, cells stably expressing CD63-Nluc were cultured in 24-well plates until reaching approximately 80% confluence. All subsequent procedures were performed at 4 °C. Conditioned medium (200 µl) was collected from the appropriate wells and transferred to microcentrifuge tubes. The tubes were subjected to centrifugation at 1000×$g$ for 15 min to remove intact cells, followed by an additional centrifugation at 10,000×$g$ for 15 min to eliminate cellular debris. Supernatant fractions (50 µl) were used for measuring CD63-Nluc exosome luminescence. Cells were kept on ice and washed once with cold PBS, and then lysed in 200 µl of PBS containing 1% TX-100 and protease inhibitor cocktail.

For the measurement of CD63-Nluc exosome secretion, a master mix was prepared by diluting the Extracellular NanoLuc Inhibitor at a 1:1000 ratio and the NanoBRET Nano-Glo Substrate at a 1:333 ratio in PBS (Promega, Madison, WI, USA). Aliquots of the Nluc substrate/inhibitor master mix (100 µl) were added to 50 µl of the supernatant fraction obtained from the medium-speed centrifugation. The mixture was briefly vortexed, and luminescence was measured using a Promega GlowMax 20/20 Luminometer (Promega, Madison, WI, USA). Following luminescence measurements, 1.5 µl of 10% TX-100 was added to each reaction tube to achieve a final concentration of 0.1% TX-100. Samples were vortexed briefly, and luminescence was measured again. For intracellular normalization, the luminescence of 50 µl of cell lysate was measured using the Nano-Glo Luciferase Assay kit (Promega, Madison, WI, USA) following the manufacturer's instructions. The exosome production index (EPI) for each sample was calculated using the formula: EPI = ([medium] - [medium +0.1% TX-100])/cell lysate.

## Acknowledgements

We thank Dr. Samantha Lewis for advice about localization to mitochondria and for sharing a plasmid; thanks Matthew J Shurtleff, David Melville, Shenjie Wu, Jordan Ngo, Congyan Zhang, Justin Williams, Morayma M Temoche-Diaz for suggestions and reading and editing the manuscript. We also thank staff at the UC Berkeley shared facilities, the Cell Culture Facility, the Flow Cytometry Facility and QB3-Berkeley (The California Institute for Quantitative Biosciences at UC Berkeley). LM and JS are supported as Research Associates of the HHMI. RS is an Investigator of the HHMI, a Senior fellow of the UC Berkeley Miller Institute of Science and Scientific Director of Aligning Science Across Parkinson's Disease.

## Additional information

### Funding

| Funder | Grant reference number | Author |
| --- | --- | --- |
| Howard Hughes Medical Institute | | Randy Schekman |

| Funder | Grant reference number | Author |
|--------|------------------------|--------|

The funders had no role in study design, data collection and interpretation, or the decision to submit the work for publication.

## Author contributions

Liang Ma, Conceptualization, Data curation, Formal analysis, Validation, Investigation, Visualization, Methodology, Writing – original draft, Writing – review and editing; Jasleen Singh, Data curation, Formal analysis, Methodology, Writing – original draft, Writing – review and editing; Randy Schekman, Conceptualization, Resources, Supervision, Funding acquisition, Validation, Investigation, Methodology, Writing – original draft, Project administration, Writing – review and editing

## Author ORCIDs

Liang Ma http://orcid.org/0000-0003-3227-5917
Randy Schekman http://orcid.org/0000-0001-8615-6409

## Decision letter and Author response

Decision letter https://doi.org/10.7554/eLife.85878.sa1
Author response https://doi.org/10.7554/eLife.85878.sa2

# Additional files

## Supplementary files

• MDAR checklist

## Data availability

All data generated or analyzed in this study are included in the manuscript and supporting files. Source data files have been provided for Figure 1b, Figure 1c, Figure 1f, Figure 1g, Figure 1-figure supplement 2a-c, Figure 2b, Figure 2g, Figure 2-figure supplement 1a-f, Figure 3a, Figure 3c, Figure 3e, Figure 3h, Figure 3 supplement 1b-c, Figure 4a, Figure 4b, Figure 4e, Figure 4f, Figure 4h, Figure 4i, Figure 4-figure supplement 1a-b, Figure 4-figure supplement 2a, Figure 5b.

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
