## [Editor Report]

This important study presents a novel mechanism of miRNA223 sorting into exosomes involving its storage within mitochondria, specifically by a mitochondrially localized protein YBAP1. The evidence supporting the findings is convincing and opens avenues for future studies on molecular mechanisms. This paper is a valuable addition to the cellular sorting of miRNA involving interplay with and between the organelles, interesting for miRNAs researchers, as well as cell biologists.

---

## [Decision Letter]

**Decision letter after peer review:**

Thank you for submitting the paper "Two RNA-binding proteins mediate the sorting of miR223 from mitochondria into exosomes" for consideration by *eLife*. Your article has been reviewed by 3 peer reviewers, and the evaluation has been overseen by a Reviewing Editor and a Senior Editor. The following individual involved in the review of your submission has agreed to reveal their identity: Alexandre Smirnov (Reviewer #1).

Essential revisions:

*Reviewer #1 (Recommendations for the authors):*

1. It would be helpful to provide, in Materials and methods, the sequences of all miR mutants used in this study, including those in Figure 2 —figure supplement 1. This would permit the reader to better appreciate to what extent the sequences and the structures of these transcripts could have contributed to the observed outcomes. I would also suggest enrolling a few additional miRNA mutants that preserve a similar amount of secondary structure – to clearly dissociate these contributions and better pinpoint the potential sequence motif.

2. The in vitro RNA packaging assay is a very interesting and powerful method that strongly resembles mitochondrial RNA import assays. Does the "membrane fraction" used in this assay eventually contain mitochondria? Can it affect the interpretation of the data? Additionally, it is not clear how the authors establish that exosome-related vesicles are actually formed in this assay. This point deserves a more detailed explanation.

3. I would suggest adding to Materials and methods a section about miRNA labelling and the microscopy shown in Figure 3a. The inclusion of additional controls (e.g. ATTO-labelled miR190) could be useful in assessing the specificity of the observed colocalisation with mitochondria.

4. The fractionation experiments described in Figure 3b-e would benefit from additional RNA controls. U6 snRNA is a good, well-behaved nuclear marker. It would be important to include more cytoplasmic transcripts, especially small ones, and compare their enrichment in mitoplasts with that of miR223 and the COX1 mRNA. This could provide a better idea about the degree of cytosolic contamination of these preparations.

5. Could the Parkin/CCCP treatment have induced miR223 transcription? This might have contributed to its increased intracellular levels. Were the YBX1 or YBAP1 cellular levels changed during this treatment?

6. To provide further support to the existence of a YBX1-YBAP1 complex in intact cells, the authors might consider using an in situ (e.g. FLIM-FRET or proximity ligation assay) or an in vivo (fluorescence complementation, BioID or similar) approach. Similarly, crosslinking miR223 to YBAP1 could be a way to further confirm this interaction in vivo.

7. Did the authors observe the same strong phenotype in YBAP1 KO cells as was frequently described in the literature (e.g. the already mentioned paper by Yagi et al., Nucleic Acids Res 2012)? Figure 5b could be supplemented with probing for YBX1 and a few mitochondrial marker proteins (especially those expressed from the mitochondrial genome and/or involved in the respiratory chain) to reflect the physiological state of the organelles.

A few textual suggestions:

L. 59: "density-based methods to separate".

Ll. 94-95: The sentence "The EMSA data showed…" is a bit confusing. Maybe add a comma after "miR223".

Ll. 108-109: "Figure 1g". The conclusion that CTD may facilitate the folding of CSD does not immediately follow from this experiment (CTD could instead provide an extension to the RNA-binding site and thereby boost the CSD affinity). Some extra context is required to draw such a conclusion (e.g. previous studies that excluded CTD from RNA binding).

L. 127: Given the setup of the experiment shown in Figure 2 —figure supplement 1, it would be safer to conclude that "the competitive binding of miR223mut (3-6) and miR223mut (4-7) was decreased". One cannot fully disentangle thermodynamic (affinity, Kd) and kinetic (kon) factors in these experiments: these mutants might be simply slower to bind.

Ll. 133-136: This sentence could be revised for clarity and accompanied by a reference to Figure 2b, c.

Ll. 140-141: It is not clear what "~4-fold dependent on the putative exosomal sorting motif" means. The reference should be to "Figure 2e".

Ll. 176-177: A better way to render it would be "Analysis of RNA extracted from isolated membranous organelles".

Ll. 180 and 325: The term "organization" is not quite appropriate here. Why not "localization"?

Ll. 201-203: It is difficult to understand how the experiment described in the previous paragraph resulted in the conclusion that "miR223 is enclosed within mitochondria". The described treatments support its partial association with mitochondrial membranes, but do not permit to establish whether miR223 was membrane-bound on the interior or the exterior side of the organelles.

L. 204: The experiment described in this section is not really looking at the "cellular location of miR223 within cells". It is more about the distribution between the cell and the extracellular milieu upon mitochondrial depletion.

Ll. 583-584: Could it be "hsa-miR-223-3p" and "hsa-miR-190-5p"?

L. 664: The reference "Rio et al., 2014" seems to be missing.

L. 916: "overexpressed in HEK293T cells".

L. 919: The formula "transfected with YBAP1-GFP" is potentially misleading. Were the cells transfected with a YBAP1-GFP-expressing plasmid?

Ll. 929, 953, 956: "standard deviations".

The definition of the error bars and the sample sizes are missing in some figures.

*Reviewer #2 (Recommendations for the authors):*

Some of the sentences within the manuscript are massive run-ons and take several reads to understand what is being said.

*Reviewer #3 (Recommendations for the authors):*

I detail below a few suggestions for additional experiments.

The IF images of figure 6 are only showing 2 P-bodies moving between mitochondria and enlarged endosomes. This observation does not demonstrate that such movement is instrumental in the transfer of miR223 between mitochondria and putative exosomes (Figure 6 and model in figure 7). At least repeating these experiments with a fluorescence staining of miR223 must be performed.

The experiments analysing exosomes or EVs and their content in miR223 should systematically include quantification of the actual number of vesicles released by the modified cells, to determine whether the decrease of miR223 is due to the general decrease of EVs or to the actual loading of miR in the EVs, and also with quantification of the miR190, which is not supposed to behave like miR223, both should be shown in panels of principal figures (figures 1a, 3a, 3c, 3j, 4c, 4g, 5c, 5d: miR190; figures 1a, 3i and 5a: number of vesicles, and especially of CD63-exosome enriched versus CD9 or CD81-plasma membrane-derived EV enriched, released per cell upon ko or CCCP treatment).

Also, it is unclear at several steps why the authors call the EVs they analyse "exosomes" since they seem to analyse a mixture of small EVs that float into a sucrose gradient at the interface of 10 and 40% sucrose. The authors have published several previous articles exploring the MVB association of YBX1 using different approaches, but it is unclear what criteria they use here to decide that the analysis shown is specific to exosomes. In their previous work on HEK293 (Shurtleff 2016), the authors had specifically isolated CD63-positive EVs from this mixture, and given the strong steady-state enrichment of CD63 in MVBs, had considered these vesicles as mainly consisting of exosomes. In a further article performed on another human cell line, MDA-MB-231 (Temoche-Diaz 2019), the authors performed a very resolutive density gradient, and showed the existence of separate populations of EVs, enriched in CD9 or CD63, but not both, and floating at different densities. Finally, in the most recent publication (Liu 2021), the authors observed YBX1 association with the high-density EVs containing CD63, however, I did not see a demonstration in this previous article that YBX1 was absent in the low-density CD9-bearing EVs. In the current work, the authors use the basic gradient, and both high and low-density EVs are most likely recovered in the 10/40% interface of the sucrose gradient used here. In fact, the Materials and methods use the term "extracellular vesicles" without specifying "exosomes". Thus, I would suggest that the authors justify clearly why they consider their mixed EV preparation to be exosomes, if necessary they can summarize the major results of their previous work that lead them to conclude that they focus on exosomes in the current work, and if justification is not convincing, they should instead change their nomenclature to use the generic term EVs.

Another issue is that the authors make strong conclusions on MVBs and exosomes when they only analyse artificially-enlarged endosomes induced by overexpression of mutant Rab5. Although this approach has been used previously and shown CD63 in these induced enlarged compartments, it is an artificial blocking of normal endosomal trafficking. These observations of YBX1 P-bodies must be confirmed by comparing them with actual regular CD63-positive MVBs of WT (versus YBX1ko, YBAP1 ko or CCCP-treated) cells. These immunofluorescence experiments are also not quantified, and the behaviour of YBX1 in the non-P-body locations, which could be close to the plasma membrane (labeled by e.g. cd9 or CD81) and influence the miR content of EVs budding from the plasma membrane should be analysed.

Finally, all bar graphs must be replaced by representations displaying the position of the individual biological replicates, so the reader can visualize the reproducibility of the experiments.

Other points:

In the parkin-GFP cells, is miR223 also accumulated in mitochondria, and how does the CCCP treatment affect this localization? I would have expected the disappearance (degradation) of miR223 together with the disappearance of mitochondria in the CCCP condition, but the miR223 quantification in cells suggests instead an accumulation in the cytosol: can the authors document this hypothesis by immunofluorescence as in Figure 3A? Concerning the EV/exosome release, I would expect the mitochondrial absence to globally prevent the release of EVs (because of lack of energy), rather than specifically controlling miR223 targeting EVs.

Small EVs containing mitochondrial components (mitovesicles) have been recently described in the literature (D'Acunzo, Sci Adv 2021): could such vesicles be involved in the targeting of miR223 to small EVs, rather than the P-body-mediated process suggested by the authors?

Results of Figure 2 show that the UCAGU sequence of miR223 contributes to the binding to YXB1, however, it is not "critical" (line 154) but instead only partial since the mutated miR223 and miR190sort have the equivalent binding ability and 50 times lower than that of miR223: do the authors speculate that other parts of the miR223 sequence are involved in binding to YBX1?

---

## [Author Response]

Essential revisions:Reviewer #1 (Recommendations for the authors):1. It would be helpful to provide, in Materials and methods, the sequences of all miR mutants used in this study, including those in Figure 2 —figure supplement 1. This would permit the reader to better appreciate to what extent the sequences and the structures of these transcripts could have contributed to the observed outcomes. I would also suggest enrolling a few additional miRNA mutants that preserve a similar amount of secondary structure – to clearly dissociate these contributions and better pinpoint the potential sequence motif.

The sequences of all miRNA mutants used in this study were added in figure 2—figure supplement 1.

It is difficult to reliably predict the secondary structure of short sequences such as of miRNAs. Although we cannot cleanly distinguish the effects of sequence and secondary structure influences in the interaction of YBX1 and miR223, the results of our analysis of the mutant sequence variants on sorting of wt and mutant miR223 and wt and mutant miR190 were consistent in the binding assay and in the exosome secretion of mutant vs wt miRNAs expressed in transfected cells. We have modified our manuscript in reference to the sorting signal to indicate this may be a sequence or structural feature of miR223.

2. The in vitro RNA packaging assay is a very interesting and powerful method that strongly resembles mitochondrial RNA import assays. Does the "membrane fraction" used in this assay eventually contain mitochondria? Can it affect the interpretation of the data? Additionally, it is not clear how the authors establish that exosome-related vesicles are actually formed in this assay. This point deserves a more detailed explanation.

We thank the reviewer for pointing this out. We too were concerned about this possibility. This issue was addressed in Shurtleff et al., (2016) where an independent assay of CD63-luciferase internalization in the cell-free reaction was used to assess a common requirement for the packaging of CD63 and miR223. We performed this in vitro packaging assay in the presence of an inhibitor (GW4869) of neutral sphingomyelinase (NS2). This inhibitor has been shown to reduce the secretion of exosomes and exosome-associated miRNAs in other studies (Li et al., 2013; Trajkovic et al., 2008; Yuyama et al., 2012). In our cell-free assay, GW4869 inhibited the protection of CD63-luciferase and miR-223 at concentrations known to inhibit the activity of NS2 in partially purified enzyme fractions (Shurtleff et al., 2016 Figure 3 and 4D). We concluded that our cell-free reaction provides a model that mimics aspects of exosome biogenesis. This published finding is now summarized in the Results and Discussion section.

Additionally, we attempted to import miR-223 into purified mitochondria under the same conditions, but we did not observe successful import (JS, data not shown). In conclusion, the in vitro packaging assay primarily mimics exosome-related vesicle biogenesis.

3. I would suggest adding to Materials and methods a section about miRNA labelling and the microscopy shown in Figure 3a. The inclusion of additional controls (e.g. ATTO-labelled miR190) could be useful in assessing the specificity of the observed colocalisation with mitochondria.

We appreciate the reviewer for bringing up the issue regarding the dye. We performed the localization experiment using transfected Atto-647N-labeled miR190 and observed fluorescence colocalized with mitochondria consistent with the reviewer suggestion that the Atto-647 dye itself may have a propensity for mitochondrial localization. We have replaced Figure 3a with a new experiment using mitochondrial immunoprecipitation to localize mi223. Immunoprecipitated mitochondria subjected to osmotic shock to generate mitoplasts confirmed that miR223, but not miR190 or U6, is highly enriched in mitoplasts.

4. The fractionation experiments described in Figure 3b-e would benefit from additional RNA controls. U6 snRNA is a good, well-behaved nuclear marker. It would be important to include more cytoplasmic transcripts, especially small ones, and compare their enrichment in mitoplasts with that of miR223 and the COX1 mRNA. This could provide a better idea about the degree of cytosolic contamination of these preparations.

As pointed out by the reviewer, U6 is a widely used miRNA control in many studies. In a previous study, it was demonstrated that miR-1 is enriched in mitochondria, the authors also used U6 as a small RNA control (Zhang et al., 2014). In our experiments, miR190 has been used as a cytosolic small RNA marker. In our mito-IP experiment, we did not observe enrichment of miR190 in the mitochondria or mitoplasts, which suggests the absence of cytosolic contamination in these samples. Furthermore, we did not detect the presence of tubulin or GAPDH in the mitochondria or mitoplast samples.

5. Could the Parkin/CCCP treatment have induced miR223 transcription? This might have contributed to its increased intracellular levels. Were the YBX1 or YBAP1 cellular levels changed during this treatment?

We cannot rule out the possibility that upregulation of miR223 is directly caused by CCCP treatment. Nonetheless, our data suggests that mitochondria contribute to the secretion of miR223 into exosomes. When mitochondria are removed by mitophagy, cytosolic miR223 is not efficiently secreted, which provides an alternative explanation for the observed increase in miR223 level after mitochondrial removal.

During CCCP treatment, the level of YBX1 remained unchanged, indicating that it is not affected by mitochondrial removal. On the other hand, YBAP1 was nearly eliminated upon CCCP treatment (Author respond image1). We have now included an additional result using a CD63-luciferase marker to show that CCCP treatment did not change the level of exosome secretion (Figure 3, supplement 2)

**Author response image 1. sa2fig1:** 

6. To provide further support to the existence of a YBX1-YBAP1 complex in intact cells, the authors might consider using an in situ (e.g. FLIM-FRET or proximity ligation assay) or an in vivo (fluorescence complementation, BioID or similar) approach. Similarly, crosslinking miR223 to YBAP1 could be a way to further confirm this interaction in vivo.

We utilized a split GFP assay to examine the interaction between YBX1 and YBAP1 in intact cells. U2OS cells were transfected with plasmids mito-BFP, GFP(1-10)-YBX1, and YBAP1-mCherry-GFP11. We observed a GFP signal surrounding mitochondria when GFP(1-10)-YBX1 and YBAP1-mCherry-GFP11 were co-transfected, possibly indicating the presence of a transient YBX1-YBAP1 complex in intact cells (Author response image 2).

Furthermore, a previously published study demonstrated that YBAP1 can be released from mitochondria into the cytosol during HSV infection (Song et al., 2021). This finding raises the possibility that YBX1 may interact with the released YBAP1.

7. Did the authors observe the same strong phenotype in YBAP1 KO cells as was frequently described in the literature (e.g. the already mentioned paper by Yagi et al., Nucleic Acids Res 2012)? Figure 5b could be supplemented with probing for YBX1 and a few mitochondrial marker proteins (especially those expressed from the mitochondrial genome and/or involved in the respiratory chain) to reflect the physiological state of the organelles.

We did not observe any growth or morphological differences between 293T WT and YBAP1-KO cells in our study. Both cell lines stably expressed 3-HA-GFP-OMP25, and the morphology of mitochondria in 293T YBAP1-KO cells appeared similar to that of 293T WT cells (Author response image 3).

We have now included new results where exosome secretion has been quantified, as measured by use of a Nanosight tracking device on buoyant density purified vesicles (for secretion of exosomes in wt vs. YBX1 deleted cells – Figure 1, supplement 1) or assays of extracellular CD63-luciferase (https://doi.org/10.7554/*eLife*.86556) (for secretion of exosomes in wt vs. YBAP1 deleted cells – Figure 5, supplement 1)

**Author response image 3. sa2fig3:** 

Additionally, we performed immunoblot analysis to detect YBX1 and several mitochondrial marker proteins in both cell lines (data has been included in Figure 5b). The results showed no significant differences between 293T WT and YBAP1-KO cells.

A few textual suggestions:L. 59: "density-based methods to separate".

Has been changed.

Ll. 94-95: The sentence "The EMSA data showed…" is a bit confusing. Maybe add a comma after "miR223".

A comma has been added.

Ll. 108-109: "Figure 1g". The conclusion that CTD may facilitate the folding of CSD does not immediately follow from this experiment (CTD could instead provide an extension to the RNA-binding site and thereby boost the CSD affinity). Some extra context is required to draw such a conclusion (e.g. previous studies that excluded CTD from RNA binding).

Has been changed to “The CTD of YBX1 did not appear to directly bind miR223 but may somehow facilitate a higher affinity interaction with miR223”.

L. 127: Given the setup of the experiment shown in Figure 2 —figure supplement 1, it would be safer to conclude that "the competitive binding of miR223mut (3-6) and miR223mut (4-7) was decreased". One cannot fully disentangle thermodynamic (affinity, Kd) and kinetic (kon) factors in these experiments: these mutants might be simply slower to bind.

Has been changed to "the competitive binding”.

Ll. 133-136: This sentence could be revised for clarity and accompanied by a reference to Figure 2b, c.

Kd of miR223mut and miR190sort were added in the figure 2c.

Ll. 140-141: It is not clear what "~4-fold dependent on the putative exosomal sorting motif" means. The reference should be to "Figure 2e".

The reference has been added into Figure2e.

Ll. 176-177: A better way to render it would be "Analysis of RNA extracted from isolated membranous organelles".

Has been changed to "Analysis of RNA extracted from isolated membranous organelles".

Ll. 180 and 325: The term "organization" is not quite appropriate here. Why not "localization"?

Has been changed to “localization”.

Ll. 201-203: It is difficult to understand how the experiment described in the previous paragraph resulted in the conclusion that "miR223 is enclosed within mitochondria". The described treatments support its partial association with mitochondrial membranes, but do not permit to establish whether miR223 was membrane-bound on the interior or the exterior side of the organelles.

We have performed an additional test with immunoisolated mitochondria from which mitoplasts were generated by swelling and subsequently treated with protease and RNase to remove intermembrane space proteins and exposed or loosely bound RNA. These treatments resulted in the removal of the intermembrane space protein AIF (Figure 3e) with little diminution in the retention of mitochondrial miR223 (3f). The likely explanation of this is that miR223 resides within the mitoplast but we cannot exclude the possibility that it is tightly bound to the external surface of the inner membrane.

L. 204: The experiment described in this section is not really looking at the "cellular location of miR223 within cells". It is more about the distribution between the cell and the extracellular milieu upon mitochondrial depletion.

Has changed to “We sought a test of the role of mitochondria in the secretion of miR223 in exosomes”.

Ll. 583-584: Could it be "hsa-miR-223-3p" and "hsa-miR-190-5p"?

Have been changed.

L. 664: The reference "Rio et al., 2014" seems to be missing.

Has been added.

L. 916: "overexpressed in HEK293T cells".

Has been changed.

L. 919: The formula "transfected with YBAP1-GFP" is potentially misleading. Were the cells transfected with a YBAP1-GFP-expressing plasmid?

Has been changed to “transfected with a YBAP1-GFP-expressing plasmid”.

Ll. 929, 953, 956: "standard deviations".The definition of the error bars and the sample sizes are missing in some figures.

Have been added.

Reviewer #2 (Recommendations for the authors):Some of the sentences within the manuscript are massive run-ons and take several reads to understand what is being said.

We have modified some sentences. If there are other run-ons, we would appreciate a specific request for clarification.

Reviewer #3 (Recommendations for the authors):I detail below a few suggestions for additional experiments.The IF images of figure 6 are only showing 2 P-bodies moving between mitochondria and enlarged endosomes. This observation does not demonstrate that such movement is instrumental in the transfer of miR223 between mitochondria and putative exosomes (Figure 6 and model in figure 7). At least repeating these experiments with a fluorescence staining of miR223 must be performed.

We appreciate Reviewer 1 for bringing up the issue regarding the dye. The Atto-647 dye itself may have a propensity for mitochondrial localization, thus we removed that result and replaced it with more mitochondrial fractionation studies. The imaging of miRNAs indeed presents challenges. If a suitable dye can be found to specifically label miR223 and allow its localization in the mitochondria, we will certainly consider conducting this experiment in the future.

In response to this reviewer’s comment, we have repeated the experiments in Figure 6 a, b to show a larger number of YBX1 puncta colocalized with mitochondria and have visualized and quantified the movement of such puncta between mitochondria and endosomes in a violin plot, Figure 6 c, d.

The experiments analysing exosomes or EVs and their content in miR223 should systematically include quantification of the actual number of vesicles released by the modified cells, to determine whether the decrease of miR223 is due to the general decrease of EVs or to the actual loading of miR in the EVs, and also with quantification of the miR190, which is not supposed to behave like miR223, both should be shown in panels of principal figures (figures 1a, 3a, 3c, 3j, 4c, 4g, 5c, 5d: miR190; figures 1a, 3i and 5a: number of vesicles, and especially of CD63-exosome enriched versus CD9 or CD81-plasma membrane-derived EV enriched, released per cell upon ko or CCCP treatment).

Our previous publications demonstrated that miR223 levels decreased in exosomes derived from YBX1 KO cells. However, no significant changes were observed in miR190 levels (Liu *et al.*, 2021; Shurtleff *et al.*, 2016). The repeated data has been included in Figure 1a.

The knockout of YBX1 did not significantly change exosome secretion as measured by Nanosight quantification of buoyant-density purified vesicle fractions (data has been added in Figure 1-supplement 1).

We stably expressed Nluc-CD63 in both 293T WT and YBAP1-KO cell lines to provide an assay for exosome secretion (https://doi.org/10.7554/*eLife*.86556). Our results showed that there was no significant difference in exosome production between these cell lines (data has been added in Figure 5-supplement 1).

We also generated a U2OS-Parkin expressing cell line with stable expression of Nluc-CD63. We observed no significant change in exosome secretion as assayed by extracellular CD63-N-luciferase between control cells and cells treated with CCCP (data has been added in Figure 3-supplement 2).

We also added miR190 as a control, besides, we also included intracellular level of miR223 and miR190.

Also, it is unclear at several steps why the authors call the EVs they analyse "exosomes" since they seem to analyse a mixture of small EVs that float into a sucrose gradient at the interface of 10 and 40% sucrose. The authors have published several previous articles exploring the MVB association of YBX1 using different approaches, but it is unclear what criteria they use here to decide that the analysis shown is specific to exosomes. In their previous work on HEK293 (Shurtleff 2016), the authors had specifically isolated CD63-positive EVs from this mixture, and given the strong steady-state enrichment of CD63 in MVBs, had considered these vesicles as mainly consisting of exosomes. In a further article performed on another human cell line, MDA-MB-231 (Temoche-Diaz 2019), the authors performed a very resolutive density gradient, and showed the existence of separate populations of EVs, enriched in CD9 or CD63, but not both, and floating at different densities. Finally, in the most recent publication (Liu 2021), the authors observed YBX1 association with the high-density EVs containing CD63, however, I did not see a demonstration in this previous article that YBX1 was absent in the low-density CD9-bearing EVs. In the current work, the authors use the basic gradient, and both high and low-density EVs are most likely recovered in the 10/40% interface of the sucrose gradient used here. In fact, the Materials and methods use the term "extracellular vesicles" without specifying "exosomes". Thus, I would suggest that the authors justify clearly why they consider their mixed EV preparation to be exosomes, if necessary they can summarize the major results of their previous work that lead them to conclude that they focus on exosomes in the current work, and if justification is not convincing, they should instead change their nomenclature to use the generic term EVs.

In our first paper on exosome secretion, we demonstrated that miR223 was highly selectively packaged into exosomes (Shurtleff et al., 2016). We detected the presence of miR223 at each stage of the exosome purification using a buoyant density flotation method. Furthermore, this conclusion was confirmed through CD63 antibody IP where exosomes from a 20%-40% sucrose interface were enriched ~5 fold in miR223 content. In another previous paper we showed that low buoyant density CD9-positive EVs do not contain enriched miRNAs (Temoche-Diaz *et al.*, 2019). Thus, considering our previous published work, we believe that vesicles floating to a 20/40% sucrose interface enrich exosomes containing miR223.

Another issue is that the authors make strong conclusions on MVBs and exosomes when they only analyse artificially-enlarged endosomes induced by overexpression of mutant Rab5. Although this approach has been used previously and shown CD63 in these induced enlarged compartments, it is an artificial blocking of normal endosomal trafficking. These observations of YBX1 P-bodies must be confirmed by comparing them with actual regular CD63-positive MVBs of WT (versus YBX1ko, YBAP1 ko or CCCP-treated) cells. These immunofluorescence experiments are also not quantified, and the behaviour of YBX1 in the non-P-body locations, which could be close to the plasma membrane (labeled by e.g. cd9 or CD81) and influence the miR content of EVs budding from the plasma membrane should be analysed.

In previous work, we demonstrated that Rab5(Q79L) expression leads to the formation of enlarged multivesicular endosomes containing numerous intraluminal vesicles containing CD63 and YBX1 (Liu et al., 2021). These endosomes exhibit a mixture of early and late endocytic markers, including CD63. (Wegner *et al.*, 2010).

In a previous paper, we demonstrated that miR223 is highly enriched into CD63-postive exosomes (Shurtleff *et al.*, 2016). Additionally, another study conducted in our lab showed that CD9-positive, lower buoyant density vesicles (EVs) do not selectively sort miRNAs (Temoche-Diaz *et al.*, 2019).

The quantification of YBX1 positive P-bodies in relation to mitochondria and endosomes is now included in Figure 6.

Finally, All bar graphs must be replaced by representations displaying the position of the individual biological replicates, so the reader can visualize the reproducibility of the experiments.

Individual biological replicates have been added in all figures.

Other points:In the parkin-GFP cells, is miR223 also accumulated in mitochondria, and how does the CCCP treatment affect this localization? I would have expected the disappearance (degradation) of miR223 together with the disappearance of mitochondria in the CCCP condition, but the miR223 quantification in cells suggests instead an accumulation in the cytosol: can the authors document this hypothesis by immunofluorescence as in Figure 3A? Concerning the EV/exosome release, I would expect the mitochondrial absence to globally prevent the release of EVs (because of lack of energy), rather than specifically controlling miR223 targeting EVs.

In U2OS cells stably expressing Parkin-GFP, miR223 was also enriched in mitochondria (Author response 4).

**Author response image 4. sa2fig4:** 

We have observed that the levels of miR223 increase upon the removal of mitochondria using CCCP treatment. There are two potential explanations for this data. Firstly, it is possible that CCCP treatment directly increases the expression level of miR223. Secondly, when mitochondria are eliminated through mitophagy, miR223 may become resistant to degradation and consequently, not efficiently secreted. Both possibilities could account for the observed increase in miR223 levels after mitochondrial removal. A distinction between these two possibilities requires further investigation.We generated a U2OS-Parkin expressing cell line with stable expression of Nluc-CD63. We observed no significant change in the number of exosomes between the control cells and cells treated with CCCP (data has been included in Figure 3-supplement 2).

Small EVs containing mitochondrial components (mitovesicles) have been recently described in the literature (D'Acunzo, Sci Adv 2021): could such vesicles be involved in the targeting of miR223 to small EVs, rather than the P-body-mediated process suggested by the authors?

Exosomes were isolated from 293T WT and YBAP1-KO cells using a flotation method. Immunoblot analysis revealed high enrichment of exosomal markers CD63, Alix, and Tsg101 in the purified exosome samples. However, the presence of mitovesicle markers STX17 and Tim23 could not be detected in the exosome samples obtained from either 293T WT or YBAP1-KO cells (Author response image 5).

Mitovesicles may be derived from mitochondria and possess a higher density compared to exosomes. Our use of a buoyant density fractionation may have resolved exosomes from mitovesicles.

**Author response image 5. sa2fig5:** 

Results of Figure 2 show that the UCAGU sequence of miR223 contributes to the binding to YXB1, however, it is not "critical" (line 154) but instead only partial since the mutated miR223 and miR190sort have the equivalent binding ability and 50 times lower than that of miR223: do the authors speculate that other parts of the miR223 sequence are involved in binding to YBX1?

We conclude that the UCAGU sequence may represent a sorting or structural feature of miR223 responsible for stable interaction with YBX1. Substitution of this sequence reduced interaction with YBX1 27 fold. The reviewer is correct that the difference in YBX1 binding to mutant miR223 and miR190sort is small and possibly insignificant however, the addition of the “sort” sequence to miR190 substantially increased interaction with YBX1. We have now added discussion to indicate that other sequence or structural features of miR223 and miR190 may reinforce or interfere with YBX1 binding.